# All You Need is One: Capsule Prompt Tuning with a Single Vector

**Yiyang Liu[1]    James C. Liang[2]    Heng Fan[3]    Wenhao Yang[4]    Yiming Cui[5]**
**Xiaotian Han[6]    Lifu Huang[7]    Dongfang Liu[8]    Qifan Wang[9]    Cheng Han[1]***

[1]University of Missouri-Kansas City    [2]U.S. Naval Research Laboratory    [3]University of North Texas
[4]Lamar University    [5]ByteDance    [6]Case Western Reserve University
[7]University of California, Davis    [8]Rochester Institute of Technology    [9]Meta AI

## Abstract

Prompt-based learning has emerged as a parameter-efficient finetuning (PEFT) approach to facilitate Large Language Model (LLM) adaptation to downstream tasks by conditioning generation with task-aware guidance. Despite its successes, current prompt-based learning methods heavily rely on laborious grid searching for optimal prompt length and typically require considerable number of prompts, introducing additional computational burden. Worse yet, our pioneer findings indicate that the task-aware prompt design is inherently limited by its absence of instance-aware information, leading to a subtle attention interplay with the input sequence. In contrast, simply incorporating instance-aware information as a part of the guidance can enhance the prompt-tuned model performance without additional fine-tuning. Moreover, we find an interesting phenomenon, namely "attention anchor," that incorporating instance-aware tokens at the earliest position of the sequence can successfully preserve strong attention to critical structural information and exhibit more active attention interaction with all input tokens. In light of our observation, we introduce Capsule Prompt-Tuning (CaPT), an efficient and effective solution that leverages off-the-shelf, informative instance semantics into prompt-based learning. Our approach innovatively integrates both instance-aware and task-aware information in a nearly parameter-free manner (*i.e.*, one single capsule prompt). Empirical results demonstrate that our method can exhibit superior performance across various language tasks (*e.g.*, 84.03% average accuracy on T5-Large), serving as an "attention anchor," while enjoying high parameter efficiency (*e.g.*, 0.003% of model parameters on Llama3.2-1B).

## 1 Introduction

Large Language Models (LLMs) [1, 2, 3, 4] initiate a revolutionary transformation in both artificial intelligence and various domains of human activity. Among their fruitful applications, a widely adopted paradigm for adapting pre-trained LLMs to downstream tasks is known as "pretrain-then-finetune." As these LLMs continue to grow in size and complexity for enhanced performance, parameter-efficient fine-tuning (PEFT) methods [5, 6, 7, 8] have emerged as compelling alternatives to full fine-tuning, demonstrating competitive performance with noticeably lower parameter usage.

Prompt-based Learning [5, 9, 10, 11, 12, 13], a well-recognized approach within PEFT, offers a simple yet effective fine-tuning strategy in which researchers employ learnable soft prompts [5, 9, 11] to activate model capabilities without modifying existing parameters. Prompting methods [3, 14, 15, 16] initially relied on the manual design of discrete prompts, which often suffer from limited flexibility and require extensive effort. Therefore, trainable soft prompts utilizing continuous embeddings have emerged as the predominant strategy [11, 12, 13]. Nevertheless, current soft prompt-based learning have two main limitations:

---

*Corresponding author

***I. Limited Capability.*** Soft prompts are typically optimized to encode task-aware instructions to guide generation in an one-size-fits-all manner [11, 12, 17, 18]. Since soft prompts are integrated into the actual sequence, their strong performance suggests a potentially significant interplay with original input tokens in the view of attention [19, 20, 21]. Counterintuitively, our finding suggests that these task-specific soft prompts actually fail to exhibit strong interaction with input tokens (*i.e.*, Fig. 1 (left)) — they primarily attend to each other (*i.e.*, the blue box), with minimal focus on critical input tokens (*i.e.*, the red box) which input sequences predominantly attend to. This observation reveals that the task-aware design of soft prompts may limit their capability to adapt to diverse input semantics, potentially constraining the effectiveness of prompt-based learning. Consequently, a natural question arises: ❶ *Can soft prompts be interactive for improved adaptation to diverse instances?*

***II. Inefficient Searching.*** Another critical limitation of soft prompt-based learning is the time-intensive grid searching for the optimal prompt length [9, 11, 12, 22, 23]. This searching generally results in a considerable number of prompts, thereby extending the sequence length of LLM and incurring additional training overhead. Even worse, recent studies have shown that these elaborately searched soft prompt tokens may still fail to effectively capture task-aware semantics for downstream tasks [24, 25], and in some cases, certain prompt tokens can even negatively impact model performance [17]. Therefore, our research question turns into: ❷ *Is it possible to eliminate the inefficient and time-consuming grid searching of prompt length while preserving informativeness?*

In response to questions ❶-❷, we investigate the possibility of incorporating off-the-shelf instance-aware information as a part of prompts to address these limitations. Our preliminary study (see §3.1) reveals the power of instance-aware information — even **one** simple, training-free integration of instance-aware token results in noticeable performance increase. Employing a compact, fixed length design can therefore eliminate the need for extensive searching on prompt length, significantly reducing the overall training time. Our observation on attention pattern further strengthen this idea. We reveal that, unlike soft prompt tokens, single instance-aware token can consistently attend to critical input tokens (*i.e.*, structural information) and be consistently and actively attended by input sequences, acting as "attention anchors." These attention anchors can be leveraged to guide model's attention toward structurally important regions of input sequences and actively propagate guidance signals into the sequence, leading to better contextual grounding.

To this end, we propose a simple yet effective prompt-based learning strategy — Capsule Prompt-Tuning (CaPT). We integrate both instance-aware information from each input sequence and task-aware information from learnable vectors to form capsuled prompts. The instance-aware information helps preserve strongly attentive interplay with input sequences, while the learnable vector encodes task-aware inductive bias as a general guidance. Notably, CaPT can operate in an almost parameter-free manner, utilizing only one single vector, which eliminates the need for time-intensive grid searching, varied lengths across different tasks, and substantial training overhead [9, 11, 12]. Experimental results demonstrate that CaPT enjoys not only training and parameter efficiency, but also state-of-the-art performance (*e.g.*, **7.5%** higher performance compared to vanilla Prompt-Tuning for T5-Large). We further confirm that our capsule prompt tokens can successfully act as "attention anchor" to establish a mutual and contextual attention pattern. We believe our work offers an innovative perspective on efficiently leveraging instance-aware information into prompt-based learning.

## 2    Related Works

**Attention in Large Language Models.** Unlike earlier architectures such as RNNs [26, 27] and CNNs [28, 29], LLMs incorporate Transformer [1, 30, 31, 32, 33, 34, 35], allowing them to capture complex dependencies and contextual nuances, thereby achieving superior performance across various tasks (*e.g.*, classification [1, 31], translation [4, 36, 37], summarization [38, 39], question answering [2, 3]). They have revolutionized the domain of natural language processing (NLP). Recently, a growing body of research [20, 21, 40, 30, 41] has investigated the success of attention mechanism from the Transformer layer via its patterns, highlighting their critical roles in model behavior and network interpretability. Research shows that these patterns can be leveraged for various purposes, such as probing the internal representation relationship and decision-making processes of the model [20, 21, 30, 42], and for leveraging attention sink to stabilize processing over infinite input sequences [40, 43]. Though promising, such attempts remain largely underexplored within prompt-based learning [18, 19, 22]. In this work, we question the sufficiency of current prompt-based learning designs and rethink then redesign it from the perspective of attention based on our observation (see §3.1).

**Parameter-Efficient Fine-Tuning.** Parameter-Efficient Fine-Tuning (PEFT) [6, 11, 44, 45] in NLP offers solutions to the computational challenges inherent in adapting LLMs to diverse tasks under the "pretrain-then-finetune" paradigm, striving to deliver performance comparable to full fine-tuning. Generally, current PEFT methods fall into three categories: *reparameterization* [6, 7], *adapter tuning* [8, 46], and *prompt-based learning* [11, 12, 17]. Among them, *reparameterization* and *adapter tuning* face two significant limitations that hinder their applications. First, they still require a substantial amount of trainable parameters — reparameterization needs enormous low-rank matrices for every targeted linear layer [6, 47], while adapters insert entire large modules throughout the model [8, 48, 49], both resulting in heavy computational costs. Second, these approaches lack flexibility, as they often necessitate customized implementations for varying model architectures and complicate task switching due to modifications made to the model structure [50, 51].

*Prompt-based learning* offers a more flexible and efficient solution with minimal input sequence adjustment, thus enabling faster adaptation [17, 22, 52]. Despite the current success, two limitations remain unaddressed: I. Soft prompts typically capture sorely the task-aware information [5, 12]. We argue that adopting the current task-aware prompt-based learning may fail to capture critical content (*i.e.*, structural information) of input sequences, ultimately impairing its capability to establish a contextual attention pattern and constraining the overall effectiveness of prompt-based learning; II. Prompt-based learning requires grid search to determine the optimal soft prompt length [22, 44], presenting a double-edged trade-off between parameter and training time efficiency (see §4.2 and Appendix §S3). In light of this view, we propose CaPT, which deliberately addresses the aforementioned issues by integrating both off-the-shelf instance semantics and task-aware guidance.

## 3 Methodology

We introduce Capsule Prompt-Tuning (CaPT), a novel prompt-based learning approach aims to enhance LLM performance and eliminate the need for time-intensive grid search. The problem and notations are defined below, drawing on the description of P-Tuning v2 (Deep Prompt-Tuning) [53], which represents one of the strongest baselines in prompt-based learning. The key findings of the effectiveness of instance-aware semantics and prompt-based learning attention are presented in §3.1, followed by the design of CaPT in §3.2. The overall framework is shown in Fig. 3.

**Preliminary for Deep Prompt-Tuning.** Given a pre-trained LLM, the objective of Deep Prompt-Tuning is to adapt the model into new task with the learning of a set of continuous embeddings P $= \{P^1, P^2, \ldots, P^N\}$ (*i.e.*, soft prompts), where $N$ denotes the number of Transformer layers and $P^i$ represents the learnable prompts in the $i$-th layer. During fine-tuning, the entire model is frozen, with the prepended learnable prompts guide the model prediction on task. Formally, the Transformer layers with prompts are defined as:

$$
\begin{aligned}
H^1 &= L_1(P^1, E) \\
H^i &= L_i(P^i, H^{i-1}) \quad i = 2, 3, \ldots, N
\end{aligned}
\tag{1}
$$

where the embeddings $E$ of the input text are initialized with the embedding layer, and $H^i$ is the contextual embeddings processed by the $i_{th}$ Transformer layer. The different colors indicate trainable and frozen parameters during fine-tuning, respectively.

### 3.1 Key Findings

**Finding I: Task-aware prompts exhibit limited interplay with input sequences.** The effectiveness of task-specific prompt-based learning is well-recognized, however, the underlying mechanism of soft prompts remains a relatively underexplored area in current research [19, 18, 42, 24, 54]. Recognizing this, we inspire by the critical role of attention in facilitating effective information flow and shaping model behavior [20, 21, 30, 55] and analyze on the attention pattern of soft prompts. This reveals a striking trend: soft prompts predominantly attend to themselves, with only subtle interaction with the rest of key input tokens. To better illustrate the point, we analyze the attention pattern over all attention heads and encoder layers on traditional Deep Prompt-Tuning. As seen in Fig. 1 (left), regular input tokens tend to exhibit strong attention to critical positions within the input sequences (*e.g.*, the $5_{th}$ - $7_{th}$ tokens "sentence", "1" and ":"). These tokens carry key structural information that supports both syntactic and semantic parsing, enabling the model to accurately interpret the hierarchical and contextual relationships within the input [20, 56, 57]. In contrast, soft prompts predominantly attend to one another and struggle to form interactive attention links with input tokens, as they are discarded after each layer, often leaving them detached from key structural information. This observation

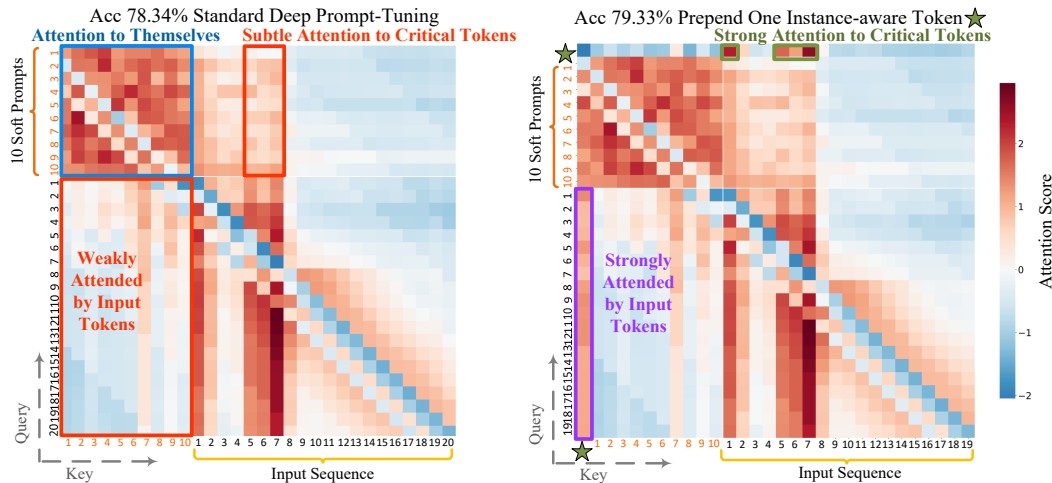

Figure 1: **Attention Analysis on T5-Base.** Attention patterns are analyzed averagely across all heads and encoder layers on the RTE validation set. Darker red indicates higher attention scores, while darker blue means lower scores. The left and right figures indicate the patterns of T5-Base with 10 soft prompts and T5-Base with one additional prepended instance-aware token (*i.e.*, ★) alongside 10 soft prompts, respectively. Critical input tokens refer to structurally important tokens such as the token "_" at $1_{st}$ and the $5_{th}$ - $7_{th}$ tokens "sentence", "1" and ":". A more detailed analysis of impact on specific heads is provided in Appendix §S5.

suggests that the current design of soft prompts is inherently limited in their ability to interact with input content like regular input tokens. Therefore, in this work, we investigate whether incorporating meaningful, instance-aware semantics as a part of prompt can foster better attentive interaction with input sequences and, in turn, enhance the performance of prompt-based learning.

**Finding II: Instance-aware tokens can enhance model performance without fine-tuning.**

We first investigate the overall significance of instance-aware semantics by figuring out whether it is effective in enhancing LLM performance. Surprisingly, we observe that even the simplest and most direct incorporation of instance-aware information yields measurable improvements across all test examples without any additional fine-tuning. Specifically, we conduct our preliminary experiments on T5-Base (see Fig. 2), compressing input tokens into short sequences of varying lengths (*i.e.*, 1, 2, 3, 4, and 10) via pooling the input sequences. These sequences are then prepended as special instance-aware tokens before the soft prompts at each layer. The results show that one single instance-aware token is able to effectively improve the test accuracy on both RTE and COPA (*i.e.*, the orange box). This suggests that incorporating instance-aware semantics, even from

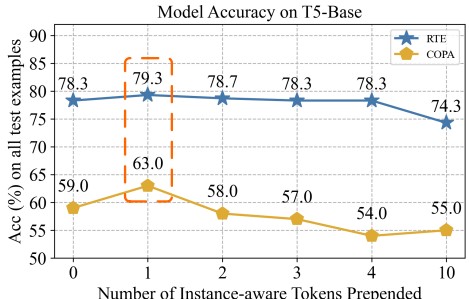

Figure 2: **Incorporating of instance-aware semantics** lifts the model performance during testing without any additional fine-tuning. (*0 token indicates the original accuracy with standard soft prompts only.*)

a minimally intrusive perspective, enhances the the overall quality of guidance signals. We also find that increasing the number of prepended instance-aware tokens (*e.g.*, 2, 3, 4, and 10) gradually declines the performance, with accuracy dropping below that of standard Deep Prompt-Tuning. We thus acknowledge that the effectiveness of guidance signals does not depend on the quantity of information provided, but rather on how well it aligns with the model's capacity to utilize it — an observation consistent with prior findings on prompt pruning strategies [5, 17]. This observation is further supported by the study on CaPT length (see §4.5).

**Finding III: Instance-aware tokens present strongly attentive interaction with input sequences as "attention anchor."** Recognizing the surprising performance gains from **Finding II**, we are excited to explore the underlying attention behavior that may explain this effect. Our analysis reveals a marked contrast in how attention is distributed across the input sequence when comparing instance-aware token to task-aware soft prompts. Unlike soft prompts, the instance-aware token successfully receives

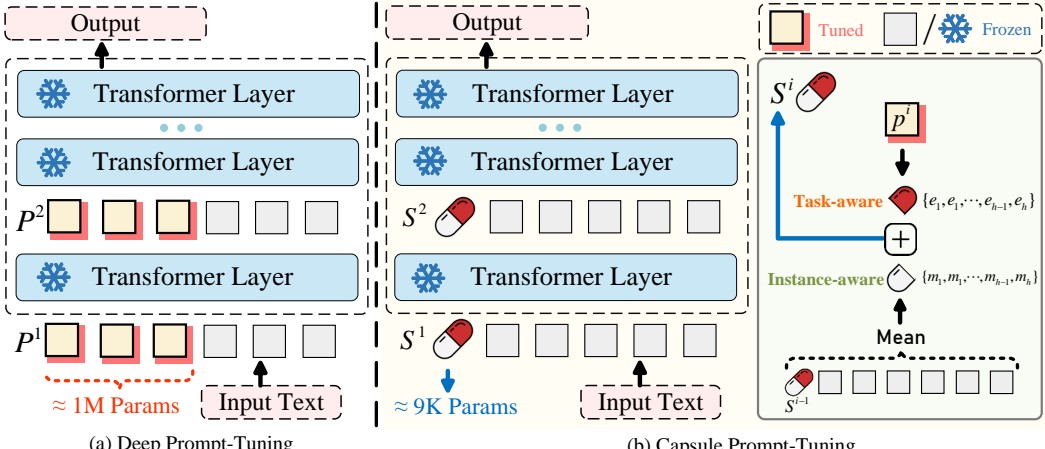

Figure 3: **Overview of Deep Prompt-Tuning** *vs.* **CaPT (ours) Frameworks.** (a) Original Deep Prompt-Tuning. (b) The overall architecture of our proposed CaPT (see §3.2), integrating both task-aware guidance and instance-aware signal to trigger "attention anchor" (see §4.3).

positive attention from input sequences (*i.e.*, the purple box), as illustrated in Fig. 1 (right). This behavior indicates that guidance signals are actively propagated into the input sequences. Moreover, the instance-aware token also consistently directs attention toward key structural tokens of the input (*i.e.*, the green box), a property that could be exploited to improve representation learning and gradient flow during fine-tuning. We refer to this phenomenon as "attention anchor," highlighting the role of instance-aware semantics in establishing stronger attentive interaction between guidance signals and input sequences. Motivated by this insight, we propose an interactive and adaptive prompt design that incorporates instance-aware information. In §3.2, we introduce the most effective and efficient design to trigger "attention anchor." Furthermore, in §4.5, we demonstrate that realizing attention anchors is highly flexible and consistently yields robust performance gains across different settings. For completeness, we include a case study examining the impact of incorporating instance-aware information on particular heads of a single instance in Appendix §S5.

### 3.2 Capsule Prompt-Tuning

Building on these observations, we propose Capsule Prompt-Tuning (CaPT), a lightweight, instance-adaptive prompt-tuning framework that eliminates costly prompt-length search while strengthening the interaction between guidance tokens and the input sequence. As shown in Fig. 3, conventional prompt-based learning prepends a fixed number of randomly initialized continuous vectors ("soft prompts") to every Transformer layer. These vectors encode task-level priors but neither adapt to individual examples nor scale gracefully across tasks, forcing practitioners to tune the prompt length by grid search. In contrast, CaPT employs a single continuous "capsule" vector per layer to compactly incorporate instance-specific information. This capsule prompt serves as a concise carrier of guidance for the model at that layer, allowing the prompt to adapt to each input instance without introducing numerous prompt parameters, achieving parameter efficiency and strong contextual grounding.

Formally, at each Transformer layer $i$, we introduce a learnable capsule $p^i$. During processing, the capsule prompt $S^i$ for layer $i$ is derived from $p^i$ in combination with a mean representation derived from the model's inputs or intermediate states, and is supplied to the Transformer alongside the usual sequence embeddings. For the first layer, $S^1$ is constructed from $p^1$ and the mean of the input embeddings $E$, and the Transformer layer $L_1$ processes this prompt together with $E$ to produce the processed capsule prompt $\underline{S}^1$ and sequence representation $H^1$. For each subsequent layer $i \geq 2$, the prompt $S^i$ is formed by combining $p^i$ with the mean of the previous processed capsule prompt $\underline{S}^{i-1}$ and sequence representation $H^{i-1}$. The layer $L_i$ then consumes $S^i$ and $H^{i-1}$, yielding $\underline{S}^i$ and $H^i$. In sum, we have:

$$
\begin{aligned}
S^1 &= p^1 + \texttt{Mean}(E) \\
\underline{S}^1, H^1 &= L_1(S^1, E) \\
S^i &= p^i + \texttt{Mean}(\underline{S}^{i-1} \oplus H^{i-1}) \quad i = 2, 3, \ldots, N \\
\underline{S}^i, H^i &= L_i(S^i,\ H^{i-1}) \quad i = 2, 3, \ldots, N
\end{aligned} \tag{2}
$$

where $S^i$ denotes the integrated capsule prompt at layer $i$ encoding instance-specific semantics. $\oplus$ denotes concatenation and $H^{i-1}$ is the representation of the input sequence from layer $i-1$.

The manner in which $S^i$ injects instance-aware semantics is flexible – for example, the capsule vector can be **_prepended_** as an extra token or **_added_** into the hidden state. In other words, $S^i$ is formed by adding the trainable capsule vector to the average embedding of the previous layer's outputs (including the previous capsule). This mean-based construction yields a compact, length-invariant prompt at each layer that adapts dynamically to the input's evolving representations through the network. Rather than relying on a large number of soft prompt tokens, we focus on a single, concise, and informative prompt vector, capturing the essential task and instance information. By prioritizing prompt quality over quantity in this way, CaPT effectively addresses the limitations of conventional soft prompting, enabling robust performance without burdensome prompt-tuning overhead.

It is worth noticing that each prompt token is shaped by both the instance semantics and one single parameter-efficient learnable vector. Specifically, the instance semantics provide semantically rich, instance-aware context to trigger "attention anchor," while the learnable vector encodes task-aware inductive bias. This design enables adaptive alignment between instance information and task objectives, ensuring that the semantic guidance remains both compact and informative. Our analysis of CaPT attention (see §4.3) confirms that the capsuled instance semantics can actively guide the model's attention toward critical input features. This guidance, as a result, promotes stronger contextual alignment and enhances task performance across diverse benchmarks (see §4.2).

## 4 Experiments

### 4.1 Experimental Setup

**Datasets.** Following common practices [11, 23], we evaluate our approach on six Natural language understanding (NLU) corpora from the SuperGLUE dataset [58], including Question Answering task (*i.e.*, BoolQ and MRC), Natural Language Inference task (*i.e.*, CB and RTE), Sentence Completion task (*i.e.*, COPA), and Word Sense Disambiguation task (*i.e.*, WiC). Since the official test sets are not publicly available, we follow [23, 59] to divide the train sets into train and validation sets by 90%/10% proportion, and translate each SuperGLUE corpus into text-to-text format. The original validation sets are considered as the test sets. More details are shown in Appendix §S1.

**Baselines.** For fair comparison, we compare our method with standard Fine-Tuning, Classification Head adaptation (*i.e.*, only tuning the linear classification head), vanilla prompt-based learning, and several state-of-the-art prompt-based learning approaches. More results are shown in Appendix §S4.

**Implementation Details.** Our method is built upon four different pretrained LLMs: T5-Base (220M) [4], T5-Large (770M) [4], Llama-3.2 (1B) [60], and Qwen-2.5 (1.5B) [61]. Specifically, we train our model under different learning rate settings for 50 epochs with early stopping, using a batch size ranging from 16 to 32. Benefiting from the simplicity and effectiveness of our design, our method does not require any hyperparameter searching which is typically time-consuming for most prompt-based learning approaches. We confirm this advantage of our design by exploring different prompt length settings in §4.5. Detailed implementation setups are provided in §S2.

**Reproducibility.** CaPT is implemented in Pytorch [62]. Experiments are conducted on NVIDIA RTX 6000 Ada 48GB GPUs. Our full implementation is available at `https://github.com/comeandcode/CaPT`.

### 4.2 Main Results

In Table 1, we report a comprehensive comparison of CaPT with other strong baselines on six NLU tasks, resulting in two key observations. ① **_Robust superior performance._** CaPT demonstrates consistently superior performance across both encoder-decoder and decoder-only architectures. On encoder-decoder T5 models, CaPT narrows the performance gap with full fine-tuning significantly, achieving **99.56%** of the average full fine-tuning performance on T5-Base. Remarkably, on T5-Large, CaPT not only outperforms strong baselines such as P-Tuning v2 and XPrompt but also exceeds the performance of full fine-tuning by 0.43%. Moreover, on decoder-only causal models Llama3.2-1B and Qwen2.5-1.5B, CaPT presents a significant improvement of **24.44%** and **10.56%** compared to Linear Head adaptation and P-Tuning v2, respectively. This consistency demonstrates the effectiveness of the attention anchor role of capsule prompt on models with different architectures. ② **_Extreme parameter and training efficient._** CaPT benefits from its almost parameter-free design,

Table 1: **Evaluation on SuperGLUE Validation Sets.** The best performance except full fine-tuning is in **bold**, and the second best is shown in underline. "*" and "†" indicate the results reported from [63] or its corresponding paper, respectively. For tasks with two metrics, the average score is reported. All scores are averaged over 3 runs.

| Method | # Para | Boolq Acc | CB F1/Acc | COPA Acc | MRC F1a | RTE Acc | WiC Acc | Average Score |
|---|---|---|---|---|---|---|---|---|
| **T5-Base** (220M) | | | | | | | | |
| Fine-Tuning† [64] | 100% | 82.30 | 91.30 | 60.00 | 79.70 | 84.50 | 69.30 | 77.85 |
| Prompt-Tuning*[EMNLP21] [11] | 0.06% | 78.12 | 84.42 | 54.37 | 78.30 | 75.27 | 62.29 | 72.13 |
| P-Tuning v2*[ACL22] [53] | 0.53% | **80.81** | 90.23 | 61.28 | 79.83 | **81.98** | 67.56 | 76.94 |
| XPrompt*[EMNLP22] [17] | 0.04% | 79.67 | 86.72 | 56.95 | 78.57 | 78.29 | 64.31 | 74.09 |
| ResPrompt*[ACL23] [52] | 0.21% | 79.25 | 85.33 | 58.64 | 78.42 | 77.14 | 62.36 | 73.52 |
| SMoP†[EMNLP23] [23] | 8e-3% | 79.40 | 86.42 | 58.30 | 79.60 | 77.50 | 65.20 | 74.40 |
| SuperPos-Prompt†[NeurIPS24] [65] | - | 74.00 | 80.20 | 62.00 | 72.90 | 70.40 | 67.60 | 71.18 |
| VFPT[NeurIPS24] [10] | 0.21% | 78.38 | 90.92 | 61.76 | 78.73 | 76.90 | 65.36 | 75.34 |
| DePT†[ICLR24] [66] | - | 79.30 | - | - | 74.30 | 79.10 | **68.70** | - |
| EPT[NAACL25] [67] | 0.06% | 79.14 | 90.18 | 56.33 | 73.43 | 78.99 | 67.71 | 74.30 |
| **Ours** | 4e-3% | 79.54 | **94.16** | **64.33** | **80.46** | 79.78 | 66.77 | **77.51** |
| **T5-Large** (770M) | | | | | | | | |
| Fine-Tuning [64] | 100% | 85.75 | 95.26 | 76.00 | 84.41 | 88.05 | 72.11 | 83.60 |
| Prompt-Tuning[EMNLP21][11] | 0.04% | 83.20 | 90.32 | 57.50 | 83.10 | 86.11 | 68.74 | 78.16 |
| P-Tuning v2[ACL22] [53] | 0.52% | **85.82** | 95.56 | 77.00 | 84.07 | **89.25** | 71.03 | 83.79 |
| XPrompt*[EMNLP22] [17] | 0.02% | 83.82 | 91.39 | 82.05 | 81.26 | 87.72 | **73.51** | 83.29 |
| ResPrompt*[ACL23] [52] | 0.15% | 83.51 | 90.64 | **82.79** | 84.02 | 86.97 | 71.13 | 83.18 |
| SMoP[EMNLP23] [23] | 3e-3% | 83.45 | 92.37 | 71.00 | 83.92 | 87.70 | 68.60 | 81.17 |
| VFPT[NeurIPS24] [10] | 0.18% | 83.89 | 93.71 | 75.63 | 83.24 | 88.10 | 71.00 | 82.56 |
| EPT[NAACL25] [67] | 0.04% | 84.77 | 93.40 | 54.00 | 80.03 | 86.33 | 71.79 | 78.39 |
| **Ours** | 3e-3% | 84.56 | **97.22** | 80.00 | **84.53** | 88.45 | 69.44 | **84.03** |
| **Llama3.2-1B** | | | | | | | | |
| Linear Head [60] | 3e-4% | 59.85 | 51.69 | 56.33 | 48.94 | 55.23 | 53.45 | 54.25 |
| Prompt-Tuning[EMNLP21] [11] | 0.06% | 60.95 | 61.61 | 57.67 | 57.00 | 62.50 | 54.70 | 59.07 |
| P-Tuning v2[ACL22] [53] | 0.53% | 62.48 | 64.29 | **61.00** | 60.34 | 58.12 | 60.15 | 61.06 |
| SMoP[EMNLP23] [23] | 0.04% | 61.13 | 62.50 | 59.33 | 57.46 | 57.40 | 54.23 | 57.51 |
| VFPT[NeurIPS24] [10] | 0.17% | 62.44 | 61.72 | 59.67 | 58.41 | 64.35 | 57.60 | 60.70 |
| EPT[NAACL25] [67] | 0.06% | 61.56 | 65.22 | 56.00 | 60.18 | 63.90 | 59.45 | 61.05 |
| **Ours** | 3e-3% | **77.28** | **65.82** | 58.00 | **65.73** | **72.56** | **65.67** | **67.51** |
| **Qwen2.5-1.5B** | | | | | | | | |
| Linear Head [60] | 2e-4% | 59.54 | 64.66 | 52.00 | 53.38 | 62.45 | 56.58 | 58.10 |
| Prompt-Tuning[EMNLP21] [11] | 0.05% | 61.38 | 65.22 | 52.33 | 53.41 | 63.18 | 56.90 | 58.74 |
| P-Tuning v2[ACL22] [53] | 0.51% | 62.08 | 68.84 | 55.33 | 56.31 | 66.43 | **59.09** | 61.35 |
| SMoP[EMNLP23] [23] | 0.03% | 61.41 | 66.76 | 54.00 | 55.34 | 64.62 | 58.15 | 60.05 |
| VFPT[NeurIPS24] [10] | 0.12% | 63.64 | 67.78 | 52.67 | 55.61 | 63.54 | 58.05 | 60.22 |
| EPT[NAACL25] [67] | 0.05% | 63.10 | 68.17 | 52.33 | 56.02 | 67.53 | 58.30 | 60.91 |
| **Ours** | 3e-3% | **64.13** | **72.42** | **57.67** | **57.49** | **68.59** | 58.46 | **63.17** |

achieving superior parameter-efficiency compared to all the other baselines (*i.e.*, $\leq$ **0.004%** parameter usage on all models). Additionally, CaPT can bypass the time-consuming grid search procedure commonly required in prompt-based learning (see §4.4), indicating the effectiveness of attention anchor in efficiently guiding model generation (see §4.3). An interesting observation is that both causal models exhibit suboptimal performance gain compared to two T5 models. We assume this is caused by the inherently different pre-training objectives and the architecture differences (*i.e.*, decoder-only *vs.* encoder-decoder), which aligned with other research's observation [68]. More analysis of CaPT length, per layer attention are conducted in §4.5 and Appendix §S6, respectively.

## 4.3 Attention Anchor

To strengthen our finding (*i.e.*, §3.1 **Finding III**) that the incorporating of instance-aware semantics can trigger "attention anchor" via enabling a mutual and context-sensitive attention, we compare both model performance and attention pattern of CaPT and Deep Prompt-Tuning with one soft prompt at each encoder layer on T5-Base. As shown in Fig. 4, we visualize the attention pattern on the RTE and CB validation sets of both models and have two key observations. *I.* Capsule prompt successfully exhibits more focused attention towards input tokens at the early positions of input sequence that carry critical structural information (*e.g.*, tokens in blue boxes are fixed structural instructions for all examples in the dataset, see Appendix §S5). This concentrated attention effectively grounds the

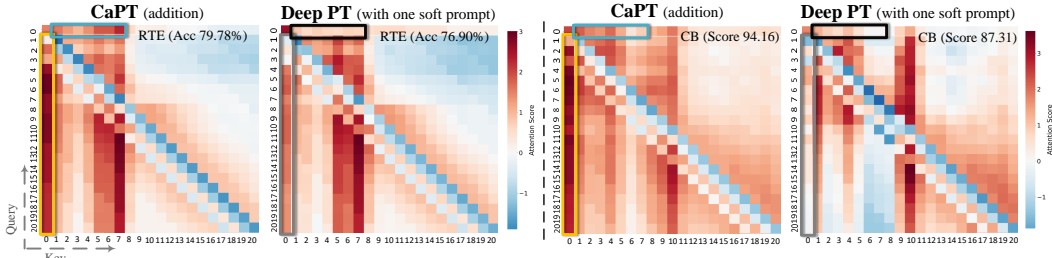

Figure 4: **Attention Analysis on T5-Base.** Attention patterns are analyzed averagely across all heads and encoder layers. Per layer encoder attention analysis and decoder attention (*i.e.*, causal attention) analysis are shown in Appendix §S6 and §S7, respectively.

LLM's focus on structural content and enhances the contextual relevance of the overall attention distribution [20, 21, 30]. In contrast, the traditional prompt struggles to attend to these key tokens (*i.e.*, black boxes), suggesting a lack of targeted interaction with structurally important tokens of input sequences. *II.* Capsule prompt receives strong and widespread attention from all input tokens (*i.e.*, yellow boxes), while the traditional prompt exhibits limited influence on input tokens (*i.e.*, gray boxes). This broad attention pattern suggests that capsule prompt provides global guidance signal that is relevant throughout the input, allowing it to play a cohesive and guiding role in generation. This observation aligns with previous studies on attention sinks [40, 69], which show that tokens designed to consistently receive attention can be able to enhance model performance. As a result, we prove that CaPT effectively triggers the "attention anchor," demonstrating a significant performance improvement compared to Deep Prompt-Tuning with a single prompt (*e.g.*, 6.73% improvement on CB) across multiple models (see Appendix §S7 for decoder-only Llama model).

### 4.4 Training Time Comparison

We compare the total training time of various prompt-based learning methods across six SuperGLUE corpora, including the searching procedures for framework-related hyperparameter (*e.g.*, prompt length, rank). For fairness, we use a consistent experimental setup: up to 50

Table 2: **Training Time Comparison on T5-Base.**

| Method | # Para | Time | Average Score |
|---|---|---|---|
| Prompt-Tuning [11] | 0.06% | 8.77× | 72.13 |
| P-Tuning v2 [53] | 0.53% | 8.37× | 76.94 |
| M-IDPG [70] | 0.47% | 12.58× | 76.96 |
| LoPA [71] | 0.44% | 14.93× | 77.98 |
| **Ours** | 4e-3% | 1.00× | 77.51 |

epochs with early stopping and a fixed batch size per task. As shown in Table 2, our default variant of CaPT demonstrates superior training efficiency, benefiting from the design that avoids grid search for prompt length optimization (see §3.2). In contrast, all baseline methods require significantly longer training durations in order to reach their optimal performances. Specifically, both Prompt-Tuning and P-Tuning v2 heavily rely on task-specific searches for optimal prompt lengths, extending the overall training time (*e.g.*, 8.77× training time). More critically, M-IDPG and LoPA involve searching for both optimal prompt length and rank, resulting in substantially higher computational overhead (*e.g.*, 14.93× training time). This observation further supports the effectiveness and efficiency of our design, which leveraging "attention anchor" to prioritize efficiency without compromising performance.

### 4.5 Ablation Study

We include a performance comparison with other PEFT methods in Appendix §S4, and analyze the differences between CaPT and other instance-incorporated prompt-based methods in Appendix §S3.

**CaPT Variants.** As stated in §3.2, the capsulation of instance-aware semantics can be achieved flexibly. Here we include three additional designs of CaPT that combines instance-aware and task-aware information, which are prepending, extraction, and projection, as shown in Fig. 5. For prepending, we prepend the mean representation of input instance to learnable task prompt as independent tokens. For extraction and projection, we consider employing learnable

Table 3: **Comparison of CaPT Variants**.

| Variant | # Para | Average Score |
|---|---|---|
| Addition | 4e-3% | 77.51% |
| Prepending | 4e-3% | 77.44% |
| Extraction | 0.03% | 77.21% |
| Projection | 0.07% | **77.64%** |

1D convolutional filters and learnable low-rank linear layers to capture instance features respectively, before integrating with the task-aware vector. The results in Table 3 shows that these variants are able

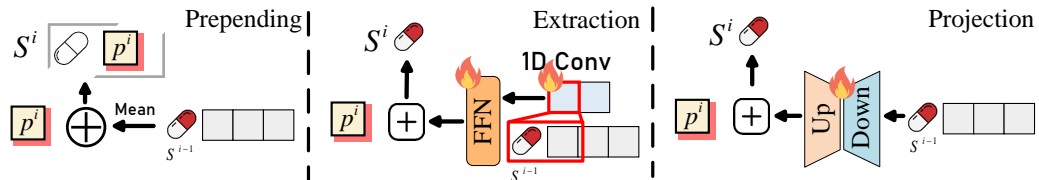

Figure 5: **CaPT Variants.** Three additional designs of CaPT are illustrated: Prepending, Extraction and Projection. The latter two designs require additional trainable modules to process hidden state (*e.g.*, 1D CNN, low-rank Linear). Of all the variants, our default addition design offers the best balance of efficiency and effectiveness, see Table 3.

to exhibit consistent performance on T5-Base. Notably, the default *addition* design (§3.2) can achieve competitive performance compared to projection (*i.e.*, 77.51% *vs.* 77.64%) with a substantial fewer parameter usage (*i.e.*, 17.5× lower). Considering the substantial reduction in time by eliminating the need for time-intensive grid search during fine-tuning, we adopt *addition* as our default method.

**CaPT Length.** In Fig. 6 (top), we explore whether increasing CaPT length is helpful for capturing quantitatively more information to exhibit a better performance on both T5-Base (220M) and Llama3.2-1B. The results reveal that employing one single capsule prompt is sufficient enough to effectively guide model adaptation. Specifically, we observe significant performance drops when prompt length increases (*e.g.*, 67.51 *vs.* 59.38), while shorter prompt lengths exhibit competitive scores compared with one single capsule prompt (*i.e.*, two capsule prompts) on both models. This observation is consist with our **Finding II** (see §3.1), indicating that the effectiveness of prompt guidance depends not on the amount of information provided, but on how well that information matches the model's ability to utilize it (*i.e.*, in our case, observed from the attention patterns).

**CaPT Depth.** We investigate the impact of applying CaPT at different sets of Transformer layers, following common practices [9, 22, 72]. Specifically, we evaluate five configurations: (a) the input layer; (b) the first half of the layers; (c) the latter half of the layers; (d) every odd-numbered layer; and (e) all layers. As illustrated in Fig. 6 (bottom), performance improves as CaPT is applied deeper in the model for both T5-Base and Llama3.2-1B. Interestingly, the configuration using every odd-numbered layer outperforms both the "first half" and "latter half" settings (*i.e.*, 75.28% *vs.* 73.67%), suggesting that sparsely distributing CaPT throughout the model may be more effective than concentrating it within a consecutive block.

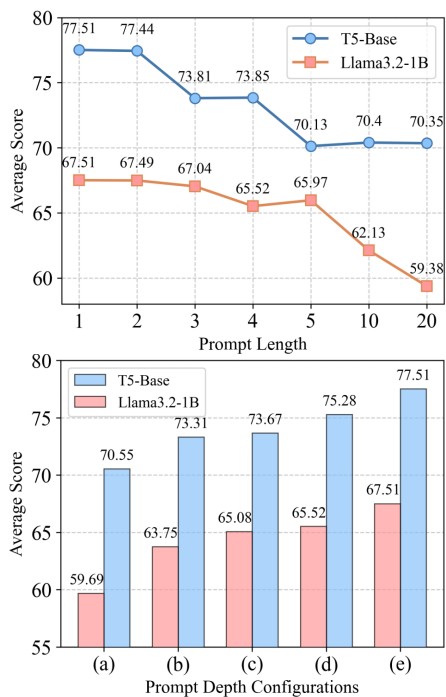

Figure 6: **CaPT Length & Depth.** The top figure shows performance across different prompt lengths, while the bottom figure illustrates the impact of prompt depth.

## 5 Conclusion

Current approaches that adapt LLMs to downstream tasks through task-aware prompt-based learning are constrained by limited interaction with input sequences and often rely on time-consuming grid searches to determine the optimal prompt length. Motivated by the significant role of instance-aware token in guiding model generation as "attention anchor," we propose CaPT — a simple yet effective framework that bridges between instance-aware and task-aware guidance signals to provide mutual and contextual attention interaction. CaPT achieves robust superior performance and eliminates the need for laborious and time-consuming prompt length searching in an almost parameter-free manner, offering an innovative perspective on LLM adaptation. We conclude that the outcomes elucidated in this paper impart essential understandings and necessitate further exploration within this realm.

## Acknowledgments

This research was supported by the National Science Foundation under Grant No. 2450068. This work used NCSA Delta GPU through allocation CIS250460 from the Advanced Cyberinfrastructure Coordination Ecosystem: Services & Support (ACCESS) program, which is supported by U.S. National Science Foundation grants No. 2138259, No. 2138286, No. 2138307, No. 2137603, and No. 2138296.

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

## S1    Dataset Statistics & Terminology

Table S1: **More details on the six SuperGLUE corpora used in our experiments.** Datasets are classified into different task categories, as suggested by [4]. NLI denotes natural language inference, QA denotes questions and answers task, SC denotes sentence completion, WSD denotes word sense disambiguation, Acc denotes accuracy, and F1a denotes the macro F1 score.

| Corpus | Examples | Task | Domain | Metric |
|--------|----------|------|--------|--------|
| Boolq | 9,427 | QA | Wikipedia | Acc |
| CB | 250 | NLI | various | F1/Acc |
| COPA | 400 | SC | blogs, encyclop | Acc |
| MRC | 27,243 | QA | various | F1a |
| RTE | 2,490 | NLI | news, Wiki | Acc |
| WiC | 5,428 | WSD | lexical databases | Acc |

Table S1 shows details of the six corpora from SuperGLUE benchmark [58] that we used for our experiments, along with their training sizes and evaluation metrics. For tasks that have two evaluation metrics, we use the average of both scores as the final performance metric following [4, 11].

We would like to further explain the important terminology used in our work:

**Task-aware** refers to information or components (*e.g.*, soft prompts) specifically designed or optimized to encode general knowledge or instructions about the dataset/task. These prompts remain the same across input instances and aim to guide the model toward task-relevant behavior.

**Instance-aware** describes information tailored to a specific input instance (*e.g.*, a sentence or document). Instead of generic task-aware instructions, instance-aware tokens reflect the feature of each individual input.

**Structurally important tokens** are the critical components of the input sequence that carry essential meaning or structure (*e.g.*, named entities, syntactic anchors). Models benefit from focusing attention on these tokens to ensure accurate comprehension and generation [20, 56, 57].

**Guidance signal** are explicit or implicit instructions (*e.g.*, hard or soft prompts) that guide models decision-making processes toward targeted tasks or behaviors.

## S2    Implementation Details

Our method employs four different pretrained LLMs: T5-Base (220M) [4], T5-Large (770M) [4], Llama-3.2 (1B) [60], and Qwen-2.5 (1.5B) [61]. Specifically, we train our models in float32 precision

Table S2: **Comparison on SuperGLUE validation sets for T5-Base.** Training time refers to the total duration required to train for up to 50 epochs with early stopping to obtain the optimal performance, using the same batch size for individual task.

| Method | # Para | Training Time | Boolq Acc | CB F1/Acc | COPA Acc | MRC F1a | RTE Acc | WiC Acc | Average Score |
|---|---|---|---|---|---|---|---|---|---|
| **T5-Base** (220M) | | | | | | | | | |
| Prompt-Tuning [11] | 0.06% | 8.77× | 78.12 | 84.42 | 54.37 | 78.30 | 75.27 | 62.29 | 72.13 |
| P-Tuning v2[53] | 0.53% | 8.37× | **80.81** | 90.23 | 61.28 | 79.83 | 81.98 | 67.56 | 76.94 |
| M-IDPG[70] | 0.47% | 12.58× | 79.60 | 92.31 | 60.33 | 79.90 | 80.90 | 68.86 | 76.96 |
| LoPA[71] | 0.44% | 14.93× | 81.09 | 91.54 | 62.00 | 80.41 | 83.40 | 69.44 | **77.98** |
| **Ours** | 4e-3% | 1.00× | 79.54 ± (0.09) | **94.16** ± (0.59) | **64.33** ± (0.33) | **80.46** ± (0.07) | 79.78 ± (0.76) | 66.77 ± (0.12) | 77.51± (0.33) |

for 50 epochs with early stopping based on validation results, using a batch size ranging from 16 to 32 to avoid memory issues. All scores are reported based on the average of three runs. For T5 models, we linearly search the best learning rate from {5, 1, 0.1}; for Llama3.2-1B, we linearly search the best learning rate from {7e-4, 5e-5, 1e-5}; for Qwen2.5-1.5B, we linearly search the best learning rate from {1e-1, 5e-3, 5e-4, 1e-5}. Benefiting from the simplicity and effectiveness of our design, our method does not require any hyperparameter searching which is typically laborious and time-consuming for most prompt-based learning approaches. This advantage is confirmed by our exploring of different prompt length settings in §4.5.

Following common practices [11, 23], we set the maximum input length, including the prompt, to 512 tokens for all experiments. Inputs that exceed this limit are truncated. We do not apply any additional preprocessing (*e.g.*, punctuation removal); instead, we directly tokenize the raw text from the SuperGLUE datasets using the appropriate tokenizer for each model. All experiments adhere to the SMoP [23] formatting, where classification tasks are reformulated into text-to-text format in T5 model. For example, in BoolQ, labels '0' and '1' are converted to 'True' and 'False,' respectively. For T5 models, we translate each SuperGLUE dataset into a text-to-text format following [23]. For Llama and Qwen models, we continue to use the previously established text-to-text template while preserving the original labels, aligning the task with LlamaForSequenceClassification and Qwen2ForSequenceClassification, respectively. All our models use the Adafactor optimizer with a linear learning rate scheduler.

## S3 Comparison with Existing Instance-incorporated prompt-based learning Approaches

While some prompt-based learning methods [70, 71, 73] have explored utilizing instance semantics, our approach distinguishes itself significantly through its simplicity, efficiency, and minimal overhead. We conduct comparison with two representative instance-based prompting methods across multiple tasks. As shown in Table S2, our method surpasses M-IDPG [70] on average performance and achieves competitive performance compared to LoPA [71] on T5-Base. While LoPA reports marginally better average performance, it requires 110× more trainable parameters (*i.e.*, 0.44% *vs.* 0.004%), which is a critical overhead under the PEFT paradigm. It requires an additional encoder (*e.g.*, CodeBert-125M [74], CodeSage-365M [75]), though frozen, still being a major reason of the significant overhead on training time. In contrast to our intuitive yet effective leveraging of "attention anchor", both methods employ heavy trainable modules (*e.g.*, MLP and multiple projectors).

## S4 Comparison with Other PEFT Paradigms

For a comprehensive evaluation of the performance of our method, we compare CaPT against other PEFT methods that differs from prompt-based learning, including Adapter [8] and LoRA [6]. While several recent reparameterization and adapter tuning methods have been proposed on different backbones and datasets, the absence of publicly released code hinders reproduction under consistent settings. As shown in Table S3, CaPT is able to surpass the other PEFT baselines (*e.g.*, 77.51% *vs.* 76.16%), while requiring a considerably fewer trainable parameters (*e.g.*, 0.004% *vs.* 1.73%). This demonstrates a unique advantage of our method, highlighting its ability to achieve superior performance with significantly reduced parameter usage.

Table S3: **Comparison with other PEFT methods on SuperGLUE for T5-Base.**

| Method | # Para | Boolq Acc | CB F1/Acc | COPA Acc | MRC F1a | RTE Acc | WiC Acc | Average Score |
|---|---|---|---|---|---|---|---|---|
| **T5-Base** (220M) | | | | | | | | |
| Adapter [8] | 0.86% | **82.50** | 88.05 | **71.50** | 75.90 | 71.90 | 67.10 | 76.16 |
| LoRA [6] | 1.73% | 81.30 | 88.20 | 70.40 | 72.60 | 75.5 | **68.30** | 76.05 |
| **Ours** | 4e-3% | 79.54 | **94.16** | 64.33 | **80.46** | **79.78** | 66.77 | **77.51** |

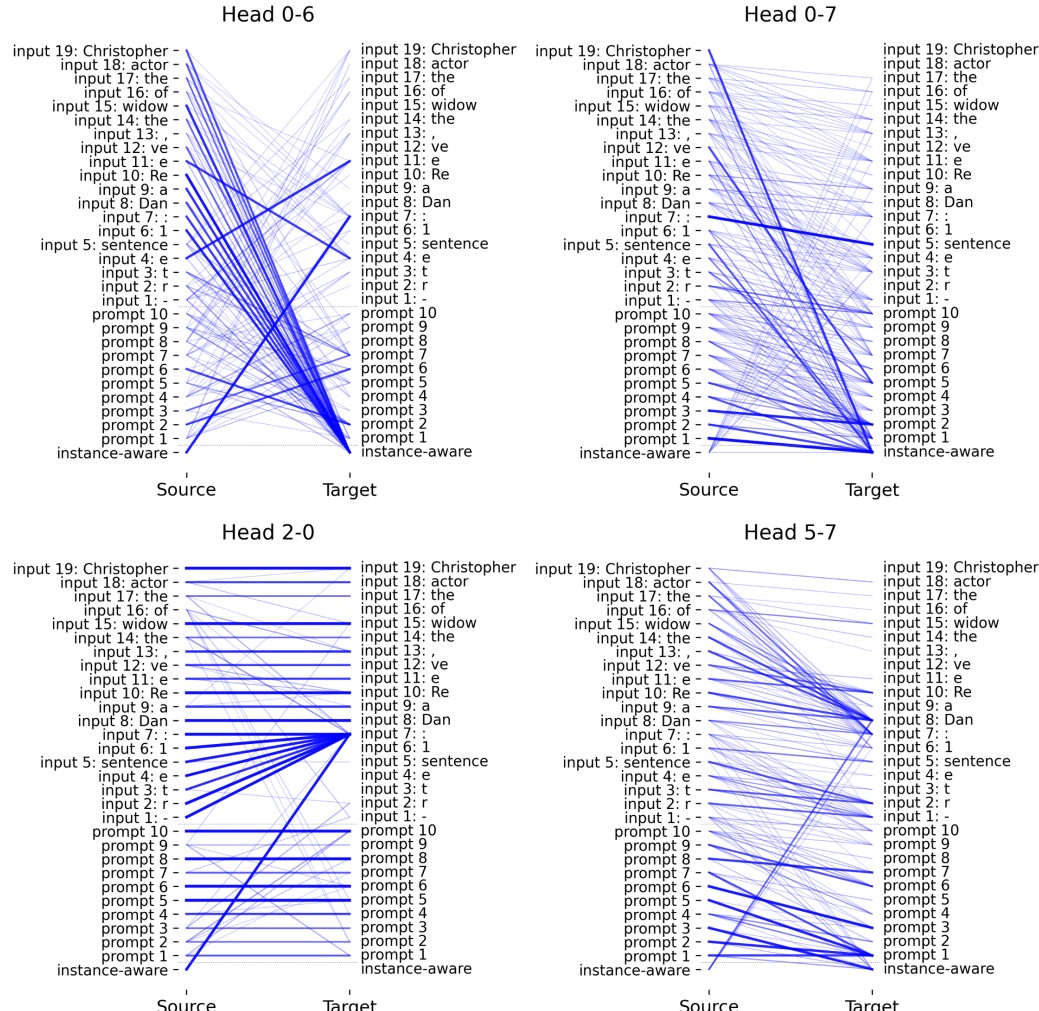

Figure S1: **Attention Heads Analysis of Incorporating Instance-aware Information.** Head a-b denotes the attention pattern of head b in encoder layer a. Darker lines represent higher attention scores. Token indices are displayed in the same format as in Fig. 1 (right).

## S5  Impact of Incorporating Instance-aware Information on Attention Heads

We present a detailed case study of **Finding III** in §3.1, showing how prepended instance-aware information affects the attention heads of the T5-Base model. Examples of heads are shown in Fig. S1. We have two key observations. First, when input tokens strongly attend to structural tokens (*e.g.*, input 7 ":" in head 2-0) and semantic tokens (*e.g.*, input 8 "Dan" in head 5-7), the instance-aware token is also able to attend to these tokens, whereas prompt tokens primarily attend to themselves. Second, the instance-aware token is effectively attended by input tokens (*e.g.*, heads 0-6 and 0-7), thereby providing guidance to the entire input sequence. These observations reinforce the role of instance-aware information as an "attention anchor," which strongly motivates our design of CaPT.

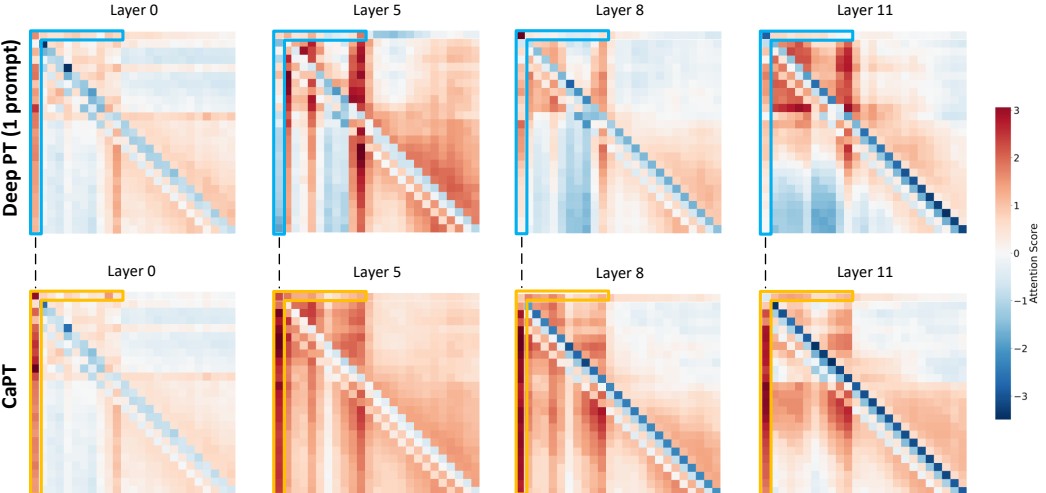

Figure S2: **Per Layer Averaged Attention on CB Validation Set for T5-Base.** Yellow regions indicate the attention behavior of capsule prompt, while blue regions show the attention behavior of traditional task-aware soft prompt.

## S6 Per Layer Attention

To further understand the "attention anchor" phenomenon, we investigate the averaged attention dynamics across different Transformer layers. Specifically, we showcase four layers (*i.e.*, layer index 0, 5, 8, and 11) of both CaPT and Deep Prompt-Tuning models in Fig. S2 to examine the significance of capsule prompt. We observe that compared to traditional deep prompt, capsule prompt is able to consistently exhibit more focused attention towards critical input tokens and strong guidance to other input tokens across all layers (*i.e.*, yellow regions) . This aligns with our averaged attention analysis across all layers in §4.3, further confirming our capsule prompt can serve as "attention anchor" to facilitate the interaction between guidance signals and input sequences. Additionally, we analyze the attention logits after applying relative positional encoding (PE) in T5, and after applying both rotary PE and causal masks in Llama. Consequently, the averaged attention scores at diagonal positions may become negative, enabling a more balanced distribution of attention toward non-self tokens, as influenced by PE mechanisms [76, 77, 78, 79, 80, 81].

## S7 Decoder-only Attention

Transformer decoders have a fundamentally different attention mechanism (*i.e.*, causal attention) compared to Transformer encoders [30, 82, 83, 84]. The paradigm of prompt-based learning on encoder-decoder T5 models focuses on applying prompts at encoder layer for better adaptation to downstream tasks [23, 17, 12, 11, 63]. Therefore, we further investigate the role of capsule prompt on decoders through decoder-only architecture (*e.g.*, Llama model). As shown in Fig. S3, even under causal attention, where tokens cannot attend to subsequent tokens, the capsule prompt is able to demonstrate superior guidance than the traditional soft prompt. This strong guidance role aligns with our observation on the T5-Base encoders. The higher performance gain (*i.e.*, 65.82 *vs.* 62.11) suggests that concentrated attention on the first token positively impacts model performance, consistent with previous studies on attention sinks in decoder-only models [40, 69]. We also find that attention scores always concentrate on the first token of the input sequence (*i.e.*, the BOS token) for both models. This phenomenon may be attributed to patterns learned during pre-training.

## S8 Impact of Task-aware and Instance-aware Guidance Signals

To understand how different forms of guidance signal contribute to performance, we separately examine task-aware (*i.e.*, Deep PT with a single prompt) and instance-aware (*i.e.*, only the mean pooled

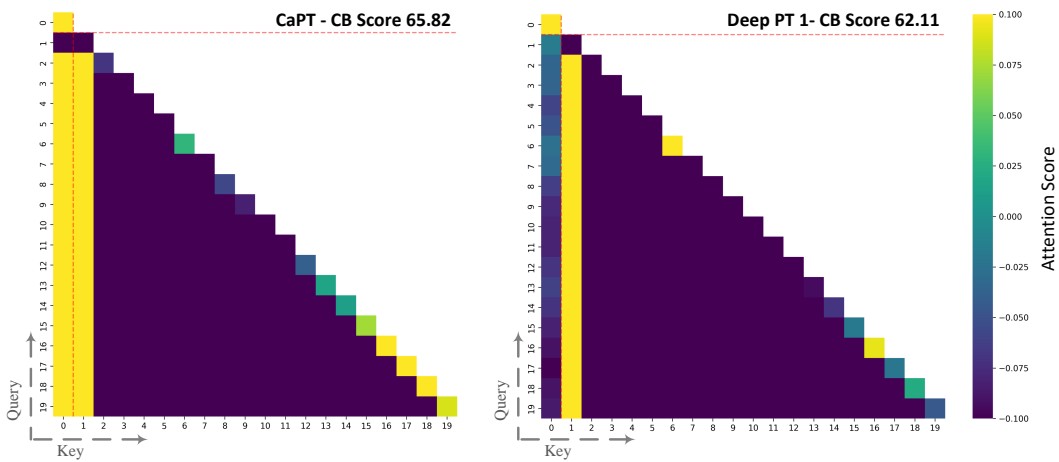

Figure S3: **Attention Analysis on Llama3.2-1B.** Attention patterns are analyzed averagely across all heads and decoder layers on CB validation set. The left figure indicates the attention pattern of CaPT, while the right figure shows the attention pattern of Deep Prompt-Tuning with one single prompt at each layer.

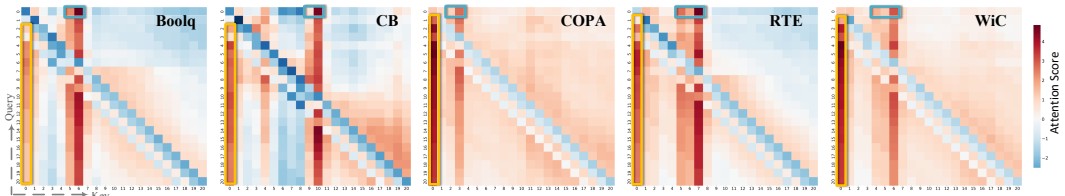

Figure S4: **Attention Patterns for the Instance-aware Guidance Only Setting.**

embedding as a prompt) guidance. As shown in Table S4, the absence of either type of guidance signal results in a performance drop, suggesting that both contribute meaningfully. Interestingly, even without any fine-tuning, instance-aware only signals can guide the model, outperforming Deep PT with a single prompt on tasks like BoolQ and WiC. Although the attention anchor effect also persists for the instance-aware only setting (see Fig. S4), CaPT achieves the best performance, validating the benefit of combining both the instance-aware and task-aware guidance.

Table S4: **Ablation study of task-aware and instance-aware guidance signals for T5-Base.**

| Method | Boolq | CB | COPA | RTE | WiC |
|---|---|---|---|---|---|
| | Acc | F1/Acc | Acc | Acc | Acc |
| **T5-Base** (220M) | | | | | |
| Insta-only-1 | 77.25 | 82.86 | 52.00 | 65.70 | 65.36 |
| Deep PT-1 | 76.51 | 87.31 | 58.00 | 76.90 | 63.80 |
| **CaPT** | **79.54** | **94.16** | **64.33** | **79.78** | **66.77** |

## S9  Further Validation of CaPT's Applicability

To further examine the applicability of CaPT, we include the comparison of P-Tuning V2 and CaPT on the CSQA [85] dataset across different models in Table S5, including results on a larger model Qwen2.5-7B. The results indicate CaPT's applicability on larger models and show that CaPT can consistently outperform P-Tuning V2. Specifically, CaPT achieves a notable performance improvement on Llama3.2-1B, highlighting the effectiveness of CaPT on CSQA.

Table S5: **Results on CSQA comparing P-Tuning V2 and CaPT across different models.**

| Method | T5-Base | T5-Large | Llama3.2-1B | Qwen2.5-7B |
|---|---|---|---|---|
| P-Tuning V2 | 56.43 | 70.52 | 45.29 | 77.72 |
| CaPT | **57.82** | **72.07** | **61.60** | **79.03** |

## S10 Ethics Concerns

CaPT is a parameter-efficient fine-tuning (PEFT) method designed to adapt pre-trained large language models (LLMs) to downstream tasks. However, it also introduces potential risks of misuse. Specifically, malicious actors may fine-tune models to generate or amplify harmful content, misinformation, or biased outputs [86, 87, 88]. To address these concerns, several mitigation strategies can be considered. These include robustness evaluations [89, 90, 91], continuous monitoring of model behavior [92], and systematic bias audits [93, 94]. Another important safeguard is the thorough documentation of models and training data, along with transparent disclosure of any known biases introduced during development [95].

## S11 Asset License and Consent

The majority of prompt tuning [53, 11], T5 [64], and Qwen2.5-1B [61] are licensed under Apache-2.0; Llama3.2 1B [60] is licensed under Llama 3.2 Community License Agreement; SuperGLUE is licensed under MIT.

All the datasets included in our study are publicly available (SuperGLUE), and all the models (T5 models, Llama3.2-1B, and Qwen2.5-1.5B) are publicly available. We would like to state that the contents in the dataset do NOT represent our views or opinions.

## S12 Reproducibility

CaPT is implemented in Pytorch [62]. Experiments are conducted on NVIDIA RTX 6000 Ada 48GB GPUs. To guarantee reproducibility, our full implementation is available at `https://github.com/comeandcode/CaPT`. Implementation details are included in Appendix §S2.

## S13 Social Impact and Limitations

This work provides a prior finding of incorporating instance-aware information as a part of guidance signals in prompt-based learning can facilitate more attentive interaction between prompts and input sequences, namely "attention anchor." Based on our finding of "attention anchor", we propose Capsule Prompt-Tuning (CaPT), an extremely lightweight, instance-adaptive prompt-based learning framework that eliminates prompt-length search while strengthening the interaction between guidance tokens and the input sequence, leading to superior performance and outstanding training efficiency. Our work is particularly beneficial for parameter-sensitive training scenarios, such as prompt tuning on resource-constrained devices and rapid adaptation with limited computational overhead.

A potential limitation of the current CaPT design is the extension to models of other modalities (*e.g.*, visual [72, 96, 97, 98]), since visual embeddings generally lack a fixed instruction template (*i.e.*, structural tokens) like those in language tasks, which the capsule prompt uses to ground more focused attention. However, given CaPT's strong guiding role, as indicated by the high attention it receives in T5 encoder layers, we believe that a carefully redesigned version of CaPT could also benefit vision tasks. In addition, while our results consistently show successful optimization of CaPT across a broad range of datasets and model scales, we acknowledge that this may not generalize to all models and datasets. Further comprehensive examinations are needed.

