# OpenReview forum: "All You Need is One: Capsule Prompt Tuning with a Single Vector"
_NeurIPS.cc/2025/Conference — NeurIPS 2025 poster_

### Official Review · Reviewer_Z96S · 2025-06-24

**Clarity:** 2
**Significance:** 3
**Originality:** 2
**Rating:** 4
**Confidence:** 4

**Summary:**

The paper proposes a simple extension to prompt tuning: append a compressed representation of the current input (emdebbings) to a soft prompt. As the paper shows, instead of appending, adding the input compression works essentially equally well, and performance is best in this case when using only a single tunable soft prompt. Overall, the result is thus a single prompt prefix vector (a capsule), that is shown to improve Q&A or classification performance on a small set of tasks and models. Interestingly, the compressed representation of the input can be obtained by simply taking the mean over all embeddings. The method compares well against other parameter-efficient fine tuning (PEFT) methods, and the paper shows some ablations and insights into the effect of capsules on average attention patterns.

**Questions:**

**Major**
1. The paper shows the combined effect of tuned single-vector prompt and mean embedding. What is missing are both contributions in isolation. As far as I understand, only tuning the single-vector prompt without the mean embedding is Fig. 4 as “DeepPT with one soft prompt”. What are the results for all experiments in Table 1 for this setting? (please point them out if they are already in the paper)
2. Same as 1. but now ablating the effect of the mean embedding only. I.e., what is the performance (Table 1) and attention matrices (Fig. 4, S2, S3) when only using the mean embedding as a capsule but without training a soft prompt? It would be particularly interesting to see whether the “attention anchor” still arises.
3. What exactly is a task in the paper? From Table X it looks like a task is identical with a dataset. If true, are the models then trained over multiple tasks, and is the soft prompt of the capsule being optimized only over a single task? Improvement: briefly describe each dataset used in the paper, by giving examples of a task and an instance.
3. How exactly are capsules trained? Please provide full details or even pseudocode in the appendix. In particular: 3a) How is it ensured that the mean embedding contains no information about the answer/class in Q&A/classification tasks? 3b) What is the objective / loss under which the soft prompt (p in Eq. 2) is trained and over which data (single task, all tasks)? 3c) Is the instance mean embedding added to the soft prompt during soft prompt training, or only at test time?
4. How can it be explained that performance goes down with capsule prompt length longer than 2? 4a) How is the mean embedding being added for longer prompt lengths (one copy per prompt vector?)?
5. If I understand correctly, the attention patterns (Figs 1, 4, S3, S4) are averaged over a single **task** (and there are multiple tasks in each dataset?). Since the crucial structural information can be marked visually per task, and not only per instance, is it correct that the same “task beneficial” attention structure could be used for the **whole task** (without making instance specific adaptations). If so, why does the task-specific soft prompt fail to do so, but the instance specific mean embedding succeeds?

**Minor comments and questions**
1. The paper uses a lot of terminology throughout, but it is sometimes not entirely clear what is meant. Please define/explain the following terms in the appendix and point the reader there early on: “task aware”, “instance aware”, “structurally important tokens”, “semantic guidance”, “attention anchor”.
2. It took me quite a while to understand the example in Fig. 1. Only after reading the appendix could I answer what is the sentence / context that the caption refers to, isn’t this a figure over *average* attention patterns where instance specific information makes no sense? Related: why are tokens 1, 5 and 7 important? What is the task and what is the instance? Please provide a short explanation and example of the task and of an instance when the figure is first discussed.
3. Fig 1b: Why are there 10 Soft prompts - I thought that there is only a single vector capsule that directly precedes the input (as shown in Fig. 3 and described in Eq. (2), similar to Fig. 4)?
4. Table 3: What is the dataset?

**Ethical Concerns:**

["NO or VERY MINOR ethics concerns only"]

**Final Justification:**

The authors have addressed my major concerns by:
* Running additional ablations (Q1, Q2).
* Clarifying their setup, in particular that all tasks have rigid structure, which means averaging attention patterns per instance reveals task structure (Q3, Q6).
* Clarifying the details of the training/tuning setup (Q4).

The authors have also provided a satisfying response / acknowledgement of my Q5.

Overall, and taking into account the other reviews and responses, my assessment of the paper remains positive, leaning more towards accept (thanks to the additional ablations), and now with increased confidence. Why not a higher score: it is still unclear how general the findings in the paper are (and there is some mild reason to be pessimistic for other kinds of tasks, see also Etjy's review), and the novelty of the main methodological improvement is somewhat limited (but certainly non-zero).

**Limitations:**

The paper repeatedly claims that the method removes the need for grid search of prompt length. I assume this is based on Fig. 2 and Fig. 6? Do these results hold for all other tasks and more importantly are there good reasons that this would hold in general? I think it would be good to add a limitation saying that based on the current findings there seems to be no need for grid search of capsule prompt length but it cannot be ruled out that this may never be the case.

**Paper Formatting Concerns:**

No concerns.

**Quality:**

3

**Strengths And Weaknesses:**

**Main contributions**
1. A simple extension to (soft) prompt tuning: add input information at test time by adding a mean embedding vector to the tune prompt.
2. Comparison of the method against other PEFT methods.
3. Illustration of the effect on average attention patterns.

**Main weaknesses**
1. The paper lacks some important details (see questions below) and the writing needs another pass for clarity.
2. The paper lacks an ablation of the effect of using only the mean embedding as a capsule (without task-specific tuning).
3. While the effect on accuracy and attention patterns is significant, it remains unclear why the mean embedding can be so efficient, and why this effect deteriorates rapidly with longer capsule prompt length.

**Verdict:**

Overall the paper presents a novel improvement over standard prompt tuning, that is very parameter efficient, and manages to effectively incorporate “instance” information in a simpler way compared to similar approaches. Through some currently unexplained coincidence, performance is best for a capsule prompt length of 1 (max. 2, but deteriorates for longer prompts), meaning a lower number of tunable parameters compared to other PEFT methods, and thus computational advantages for prompt tuning. Similarly, the capsule causes significant changes in attention patterns (compared to prompt tuning without adding the mean embedding) - which the paper hypothesizes as being responsible for the improved performance. While this is a compelling hypothesis, the current paper cannot explain why this “attention anchoring” happens (or maybe in what kinds of tasks it is expected to happen) and why it improves performance. I think an important ablation could really improve insights (see Questions below). Together with the writing that needs improvements for clarity and the somewhat limited set of tasks, I personally think that the work could be significantly improved by taking a bit more time, performing more analysis, and resubmitting elsewhere. Having said that, I currently do not think there are very strong reasons to recommend rejecting the paper, and thus lean towards a (very) weak accept. It is possible that I misunderstood parts of the setting in the paper (particularly what a task is and over which data the soft prompt is optimized over) - I have indicated this with a lower confidence score.

---

> ### Author Rebuttal · Authors · 2025-07-31
>
> We sincerely thank reviewer Z96S for the valuable feedback, which is crucial for improving our work.
>
> **Major:**
>
> **Q1.: Deep PT with one prompt.**
>
> **A1.:**
> “DeepPT with one soft prompt” in Figure 4 and S3 stands for “only tuning the single-vector prompt w/o mean embedding”. We provide a comparison isolating the contribution of the mean embedding in our response to Q2.
>
> To further validate the effectiveness of CaPT, we conduct additional experiments on Deep PT with one prompt. As seen, CaPT consistently outperforms Deep PT with one prompt, demonstrating the effectiveness of leveraging instance-aware info to serve as attention anchor.
>
> | Model           | Method     | Boolq |   CB   |  COPA |  MRC  |  RTE   |  WiC  | Average |
> |-----------------|------------|-------|--------|-------|-------|--------|-------|---------|
> | T5-Large        | Deep PT 1  | 82.81 | 94.16  | 71.33 | 83.75 | 85.92  | 66.92 | 80.82   |
> |                 | CaPT       | 84.56 | 97.22  | 80.00 | 84.53 | 88.45  | 69.44 | 84.03   |
> | Llama-3.2-1B     | Deep PT 1  | 60.06 | 62.11  | 47.33 | 54.21 | 54.15  | 55.02 | 55.48   |
> |                 | CaPT       | 77.28 | 65.82  | 58.00 | 65.73 | 72.56  | 65.67 | 67.51   |
> | Qwen-2.5-1.5B    | Deep PT 1  | 59.63 | 64.26  | 44.00 | 44.58 | 63.54  | 58.46 | 55.75   |
> |                 | CaPT       | 64.13 | 72.42  | 57.67 | 57.49 | 68.59  | 58.46 | 63.17   |
>
>
> **Q2: Regarding additional ablation.**
>
> **A2:** We conduct an ablation using only the mean pooled embedding as a capsule across all layers of the frozen T5-Base model. As seen, instance-aware only signals still guide the model effectively—outperforming Deep PT with a single prompt on tasks like BoolQ and WiC. The attention anchor effect also persists in this setting (figures to be included in revision). Notably, CaPT achieves the best performance, validating the benefit of combining instance-aware and task-aware guidance.
>
> | T5-Base     | Boolq  |   CB  |  COPA  |    RTE     |    WiC    |
> | --------------- | ---------| -------- | ---------| ----------- | ----------- |
> | Insta Only 1| 77.25  | 82.86 |  52.00 | 65.70    |   65.36   |
> | Deep PT 1  |  76.51 | 87.31 |  58.00 |  76.90   |  63.80    |
> | CaPT          | 79.54  | 94.16 | 64.33  |  79.78   |   66.77   |
>
>
>
> **Q3: Regarding tasks and datasets.**
>
> **A3:** Sorry for the confusion. We would like to further explain that a “task” refers to a corpus in SuperGLUE, such as COPA. The definition follows common practices [1, 2]. We will revise our paper to name them as “corpus” to distinguish from task categories (NLI, QA, etc) for better clarity.
>
> If you don’t mind, could you kindly clarify which table you are referring to as Table X?
>
> And yes, soft prompts of CaPT are optimized on each corpus of SuperGLUE.
>
> We have included the description of each dataset in section 4.1 and detailed statistics in Appendix S1. As for examples, an instance in the Boolq corpus is a sequence with a fixed structural instruction template: “*boolq passage:* Powdered sugar, also called confectioners' sugar ... *question:* Is confectionary sugar the same as powdered sugar?”
>
> **Q4: Implementation details.**
>
> **A4:** Each CaPT model is trained on each corpus of SuperGLUE. After careful consideration to ensure maximum clarity, we refine the Eq.(2) which is the core of constructing capsule prompts for both training and inference. There is only one training stage for each corpus.
>
> $S^1 =p^1 + Mean(E)$
>
> $\underline{S}^1, H^1 = L_1({S^1}, {E}) $
>
> $S^i = p^i + Mean(\underline{S}^{i-1} \oplus H^{i-1})  \ \ \ \ \ i = 2, 3,\dots,N $
>
> $\underline{S}^i, H^i = L_i({S^i}, \ H^{i-1})  \ \ \ \ \ i = 2, 3,\dots,N $
>
> Each capsule prompt $S^i$ is formed by adding a learnable vector $p^i$ to the mean of the previous capsule $\underline{S}^{i-1}$ and hidden state $H^{i-1}$. $S^i$ is prepended to the sequence at each layer. Vectors $p^i$ are zero-initialized and trained via gradient descent.
>
> **(a)** For T5 models, we use text-to-text format, following the common practice [2, 3]. For decoder-only models, we preserve the original labels (0, 1, 2) as classification tasks. Therefore, there is no answer in input sequences.
>
> **(b)** We use cross-entropy loss on a single corpus to update the learnable vectors. For T5, this is generative cross-entropy over the vocabulary; for decoder models, it's classification cross-entropy over class indices.
>
> **(c)** Mean embedding is added to the soft prompt during both training and inference. Capsule prompts are also constructed in the same way during both training and inference.
>
>
> **Q5: Regarding longer prompts.**
>
> **A5:**
> Our experimental results on both Finding 2 and ablation indicate that effectiveness of prompts depends not on the amount of info provided, but on how well that info matches the model’s ability to utilize it. This observation is consistent with prior works on prompt pruning [2, 4], which shows that not all prompt tokens contribute positively to performance. Moreover, studies on prompt tuning [5] and P-Tuning v2 [6] have also observed that longer prompts do not necessarily lead to better outcomes, and may even hurt performance by introducing noise or leading to slower convergence.
>
> **(a)** For longer prompt lengths, we apply adaptive mean pooling to compress embedding into multiple tokens, matching the desired prompt length instead of one copy per prompt token.
>
> **Q6: Regarding the attention pattern analysis and why standard prompts fail.**
>
> **A6:**
> **(a)** The attention patterns are averaged over a single corpus and there is one task with the same structural template in a corpus. As for the structural template for a corpus, for example, “rte sentence1: *premise* sentence2: *hypothesis*” for RTE. Since each corpus is different on data distribution, question type, and structural template, prompts are optimized on each corpus of SuperGLUE.
>
> **(b)** Standard soft prompts are randomly initialized and designed to be updated via gradient descent, but not specifically designed for interactive attention behavior. In contrast, CaPT directly carries off-the-shelf instance-aware info, resulting in a fundamental difference. Acknowledging the current research on prompt tuning performance analysis remains largely unexplored, we believe the observation of the attention anchor constitutes a foundational contribution.
>
>
> **Minor:**
>
> **Q1: Regarding terminology.**
>
> **A1:** This is a valuable suggestion! We will revise our paper accordingly. Due to the rebuttal length constraint, we would like to provide an explanation here: *Instance-aware* describes info tailored to a specific input instance (e.g., a sentence or document). Instead of generic task-aware instructions, instance-aware tokens reflect the feature of each individual input.
>
> **Q2: Regarding attention analysis.**
>
> **A2:** We would like to clarify that Fig. 1 shows the averaged attention pattern across all samples of the RTE validation set. When processing each input sample, the prepended instance-aware token is mean-pooled from each sample. We present an averaged result for a high-level observation on the general impact of instance-aware info. This averaging does not conflict with the contribution of instance-aware info. Additionally, we have also provided a concrete example in Appendix S5 which shows the attention behavior of specific heads on one specific instance.
>
> Tokens 1, 5-7 are important because they encode structural info in the input sequence—such as syntactic markers (e.g., "sentence 1:", "question:"). As shown in Fig. 1 (left), these tokens consistently receive high attention from other input tokens, indicating their role as cues that help the model interpret inputs. In prompt tuning, such structural tokens guide the model in segmenting the input into meaningful parts and in anchoring its attention, which improves task understanding and performance. This is supported by prior work showing that templated inputs with explicit structure improve few-shot learning capabilities by reinforcing input semantics [7, 8]. Additionally, prior work [9] indicates that attention heads in LLMs heavily focus on structural and syntactic info during In-Context Identification phase, aiding the model to interpret contextual cues.
>
> As suggested by Q3, we have provided a detailed explanation of tasks, datasets, and instances.
>
>
> **Q3: Explanation of Fig.1 (b)**
>
> **A3:** This is because it’s an experiment based on the prompt-tuned model in Fig. 1 (a).
>
> **Q4: Dataset for Table 3.**
>
> **A4:** The average scores are also calculated based on six SuperGLUE corpora.
>
> **Regarding the discussion on Limitation**
>
> **A:**  We thank the reviewer’s careful attention to our claim about CaPT’s ability on eliminating prompt length grid search. We will add a limitation in the revised version. While our results consistently show successful optimization of CaPT across a broad range of datasets and model scales, we acknowledge that this may not generalize to all models and datasets. During the rebuttal phase, we also extended CaPT to the CSQA, and the results reinforce the robustness of our current single-prompt design.
>
> [1] Dept: Decomposed prompt tuning for parameter-efficient fine-tuning. arXiv 2023
>
> [2] Xprompt: Exploring the extreme of prompt tuning. arXiv 2022
>
> [3] Smop: Towards efficient and effective prompt tuning with sparse mixture-of-prompts.  EMNLP 2023
>
> [4] E2vpt: An effective and efficient approach for visual prompt tuning. ICCV 2023
>
> [5] P-tuning v2: Prompt tuning can be comparable to fine-tuning universally across scales and tasks. ACL 2022
>
> [6] The power of scale for parameter-efficient prompt tuning. arXiv 2021
>
> [7] Language models are few-shot learners. NeurIPS 2020
>
> [8] Making pre-trained language models better few-shot learners. arXiv 2020
>
> [9] Attention heads of large language models: A survey. arXiv 2024
>
> We hope our response addresses your questions. Please let us know if there are any additional questions, and we will be happy to discuss further.

---

> > ### Comment · Reviewer_Z96S · 2025-08-04
> > **Thanks for the detailed response**
> >
> > Thank you for clarifying my questions and discussing my criticism. I am happy to see the additional ablations that clearly show that both parts (the mean embedding and the tuned prompt) contribute to the overall score (and thus both parts are necessary). I have no further questions, and, reading the other reviews and your responses, I remain positive and now with more confidence. Below are some comments to clarify my initial questions, which may be helpful to improve the paper such that other readers do not have the same questions. I do not expect a response, this is merely to clarify.
> >
> > * Q2 - which Table X: apologies, I meant Table S1.
> > * Q3, Q6. One important aspect that I really only understood fully after the rebuttal is that each task has a rigid structure. Accordingly, it makes sense to average over instances and visualize attention patterns, etc over a corpus. Accordingly, (average) instance-aware patterns coincide with the task's structural patterns. So there is a large overlap between task- and instance-structure, which can cause some confusion when talking about instance-information, via plots that show task-structure information (e.g., average attention-patterns over a corpus). This is fine, but maybe worth clarifying. It could also be good to discuss how the general methodology and findings would apply (or not apply) for tasks where this is not the case (as a limitation). E.g., if the task was sentiment classification from movie reviews, I would expect much more variance over the attention-patterns (i.e., task structure) across instances, and that averaging over instances would not lead to very meaningful attention patterns (and attention anchoring).
> > * Q4. Another broad concern I had (the reason I asked for training/tuning details) was that taking the mean-embedding over a sequence and adding it as a prefix could, in principle, weaken the effect of causal masking, as future information for the current instance leaks into the prefix. For some tasks and evaluations this might be a problem, and could invalidate evaluation scores. E.g., the mean embedding over the whole sequence may make next-token prediction easier for parts of the sequence (which would reduce overall log-loss). Admittedly, I don't think that a single mean-embedding vector can carry too much information, such that comparisons would be too strongly affected, but I wanted to understand whether this is possible in principle for the proposed method.

---

> > > ### Author Response · Authors · 2025-08-04
> > >
> > > We're glad our response has addressed your concerns and received your approval of our work. We sincerely appreciate your additional comments on improving our work.
> > >
> > > $\bullet$ Thank you for your excellent suggestion. Empirically, we observe that instance-aware info can help preserve attention to critical structural tokens from the analysis of the average attention pattern (e.g., Fig.1) and can also help capture semantically important tokens (e.g., Head 5-7 in Fig.S1) from the analysis of specific heads for one sample. We will revise our Finding 1 to include clearer analysis results from both aspects. We will also include the scope of current findings as part of our limitations. Moreover, in the future work, we will investigate in detail about the per-instance attention pattern within a broader range of tasks.
> > >
> > > $\bullet$ Thank you for your insightful comment. Our current design of CaPT does carry high-level information of input sequences (without labels/answers) which may not be able to directly affect the causal masking. Since our current evaluation of CaPT is based on various experiments on text-to-text generation and classification (as is common practice in the prompt tuning community), as discussed above, we will include the current scope as part of our limitations. Additionally, as suggested by your excellent suggestion, we will further examine whether introducing high-level info of input sequences as prefix affects the causal masking on causal language models as our future work.
> > >
> > > Thank you so much for your thoughtful comments, and we are always ready to respond to any further questions you may have.

---

> > > > ### Comment · Reviewer_Z96S · 2025-08-07
> > > > **Thanks for the additional comments**
> > > >
> > > > Both points are fully clarified now. I have no further comments or questions.

---

> > > > > ### Author Response · Authors · 2025-08-07
> > > > >
> > > > > We sincerely appreciate your time and effort in reviewing our paper and providing valuable feedback, which is essential for improving our work. We're glad that our responses have addressed most of your concerns and that you've maintained the positive assessment of our work.

---

### Official Review · Reviewer_Q1NL · 2025-06-27

**Clarity:** 3
**Significance:** 3
**Originality:** 2
**Rating:** 4
**Confidence:** 3

**Summary:**

Existing soft prompt-based learning methods face two main limitations: limited capability and inefficient prompt searching. To address these issues, the authors explore incorporating off-the-shelf instance-aware information into prompts. They find that even a simple, training-free instance-aware token can significantly improve performance and reduce the cost of prompt tuning. Moreover, such tokens act as "attention anchors," consistently guiding the model’s focus to structurally important regions of the input. Building on this insight, the paper proposes CaPT, which constructs capsule prompts by integrating instance-aware information from each input sequence with task-aware information from learnable vectors. Empirical results demonstrate that their method can exhibit superior performance across various language tasks.

**Questions:**

See weaknesses

**Ethical Concerns:**

["NO or VERY MINOR ethics concerns only"]

**Final Justification:**

Most of my concerns have been clarified. I will increase the score accordingly.

**Limitations:**

yes

**Paper Formatting Concerns:**

I didn't find any major formatting issues in this paper.

**Quality:**

3

**Strengths And Weaknesses:**

Strengths:

1. The paper is well-structured and logically sound. It clearly demonstrates the limitations of existing soft prompt-based methods through empirical analysis and proposes a practical solution.

2. The writing is clear and accessible, making the paper easy to follow.

3. The experimental evaluation is comprehensive, including a wide range of strong baselines and effectively validating the proposed approach.

Weaknesses:

1. The evidence for Finding 2 is limited to experiments on T5, which may raise concerns about its generalizability across architectures.

2. The proposed method for obtaining instance-aware information relies on mean pooling over input representations. While simple and effective, this may reduce the perceived novelty of the approach.

3. Figure 3 provides a parameter comparison between existing methods and CaPT. A rough analysis of space complexity would further enhance the reader’s understanding of the method’s efficiency.

---

> ### Author Rebuttal · Authors · 2025-07-31
>
> We sincerely appreciate the time and effort you've devoted to reviewing our work and providing helpful feedback!
>
>
> **Q1: Regarding Finding 2’s generalizability.**
>
> **A1:**
> Thank you for the question! To assess the generalizability of our Finding 2, we further investigate simply prepending instance-aware information on a decoder-only model (i.e.,  Llama-3.2-1B) without additional fine-tuning. As shown in the table below, the results on Llama-3.2-1B strengthen our Finding 2 that indicates the significance of instance-aware information.
>
> While CaPT is initially motivated by findings from T5, in our paper we have applied CaPT to decoder-only models (i.e., Llama-3.2-1B and Qwen-2.5-1.5B). Experimental results (see Table 1) show that CaPT can work effectively across different architectures. We have observed a similar attention anchor role of CaPT on Llama-3.2-1B (please see Appendix S7), aligning with our observation on T5-Base.
>
> | Task  | Deep PT |  Prepending one single instance-aware token |
> | ---- | ----------- | --------------- |
> | Boolq  |  62.48      |    65.14        |
> | RTE |   58.12     |      63.18      |
>
>
> **Q2: Regarding CaPT’s novelty and mean pooling design.**
>
> **A2:**
> Thank you for raising this question. In section 4.4, we have provided different instance-aware information incorporation strategies, including prepending, 1D CNN feature extraction, and feature projection. Among these different strategies, mean pooling with addition is considered as the most effective and straightforward approach, primarily for three key reasons: **First**, mean pooling is training-free and requires no additional modules to obtain instance-aware information, ensuring computational and parameter efficiency. **Second**, mean pooling with addition can demonstrate notable effectiveness with substantially fewer parameter usage (e.g., 17.5x lower than the feature projection strategy). **Third**, this design consolidates a large set of soft prompt tokens into a single prompt, thereby significantly mitigating the existing training overhead associated with searching for an optimal prompt length.
>
> We would like to further highlight several key contributions of CaPT besides its intuitive and effective design:
>
> $\bullet$ First, our work is the first attempt to analyze the bidirectional attentive interaction between soft prompts and input tokens in the NLP field. We uncover the power of off-the-shelf instance-aware information that can trigger interactive **attention anchor**.
>
> $\bullet$ Second, CaPT’s superior performance further confirms the significance of employing instance-aware information as part of guidance signal through comprehensive experiments on various language tasks.
>
> $\bullet$ Third, with the incorporation of both instance- and task- aware information in the guidance signals, CaPT eliminates the need for grid search of the optimal soft prompt length. This advantage significantly reduces the overall computational overhead (please see section 4.4).
>
> We will add additional discussions in revision to make it clearer. Thank you!
>
>
> **Q3: Regarding space complexity analysis.**
>
> **A3:**
> Thank you for the suggestion. We include a space complexity analysis on Figure 3 for a clearer comparison with Deep PT. Let $L$ denote the number of layers with prompts, and $d$ denote the embedding dimension. Given that the number of tokens for Deep PT is $n \gg 1$, the space complexity can be formally expressed as:
>
> | Methods   |  Prompt Space Complexity |
> | -------------- | ----------------------- |
> | Deep PT    | $O(L×n×d) $            |
> |  Ours         |  $O(L×1×d)=O(L×d)$ |
>
> As seen, CaPT achieves a significantly lower space complexity of $O(L×d)$, as it requires only a single prompt per layer, enabling CaPT to be more efficient with superior performance.
>
>
> We sincerely appreciate your thoughtful comments. We hope our response addresses your questions. Please let us know if there are any additional questions, and we will be happy to discuss further.

---

> > ### Comment · Reviewer_Q1NL · 2025-08-03
> >
> > Thanks for the response. I will maintain my positive rating.

---

> > > ### Author Response · Authors · 2025-08-03
> > >
> > > Thank you for maintaining your positive rating. We are always ready to respond to any further questions you may have, and we sincerely appreciate your thoughtful comments that help us improve our work.

---

### Official Review · Reviewer_Etjy · 2025-07-05

**Clarity:** 3
**Significance:** 2
**Originality:** 2
**Rating:** 4
**Confidence:** 4

**Summary:**

This paper introduces Capsule Prompt-Tuning (CaPT), a novel approach to enhance prompt-based learning for large language models (LLMs). The authors identify that existing task-aware prompts lack instance-aware information, resulting in suboptimal attention mechanisms. This method efficiently integrates both task-aware and instance-aware information using a single capsule prompt, achieving high parameter efficiency.

**Questions:**

see weakness

**Ethical Concerns:**

["NO or VERY MINOR ethics concerns only"]

**Final Justification:**

The authors have provided more results as suggested in their response,  and the results clarify my concerns, justifying the contribution of this paper. I increased the score.

**Limitations:**

yes

**Quality:**

3

**Strengths And Weaknesses:**

**Strengths:**

1. The paper provides a compelling attention analysis, showing that current task-aware prompts exhibit limited interplay with input sequences. The comparative results lend credibility to this claim.
2. The authors demonstrate the effectiveness of the proposed CaPT method. Experimental results on the LLaMA3.2-1B model show that CaPT achieves performance gains while updating only 0.003% of model parameters and requiring only 1/8.77 of the training time compared to other methods.
3. The paper is well-written. Terminology is clearly introduced, and the sections are well-structured, allowing for smooth and easy reading.


**Weaknesses:**

1. **Scope of Applicability**.
   (1) Although CaPT is evaluated on several natural language understanding tasks, these tasks primarily involve text classification and relatively simple QA settings. Its effectiveness on more complex reasoning tasks—such as mathematical or commonsense reasoning—remains uncertain.
   (2) Furthermore, the experiments are limited to relatively small models (220M, 770M, 1B, 1.5B). It is unclear whether CaPT remains effective on larger-scale LLMs (e.g., 7B or above).
   (3) According to Table 1, CaPT does not outperform all baselines across all tasks. Why is this the case? What kinds of tasks is CaPT particularly suited for?
   If the authors could further validate CaPT on tasks such as MATH500 or CSQA and larger models like 7B+, I would be more inclined to raise my score.

2. **Novelty of the Method**.
   In Section 3, CaPT is described as integrating both task-aware and instance-aware information using a single capsule prompt. Are there prior works that also attempt to combine these two types of information? If so, how does CaPT differ? If not, what are the challenges of integrating both, and how does CaPT address them? These points should be clarified more explicitly, preferably in the Introduction or Related Work section.

3. **Related Work**.
   The Related Work section could be more focused. For example, general background on large language models may not be necessary. The section would benefit from highlighting work most relevant to the core contribution of CaPT.

4. **Typoes**:

   * Equations (1) and (2) do not appear to be properly typeset, which may not meet academic presentation standards.
   * Terminology is sometimes inconsistent. For example, “capsule prompt” and “capsule vector” are used interchangeably when describing CaPT, which could confuse readers.

5. **Experimental Details**:
   The paper lacks important implementation details. For example, how is the capsule prompt obtained during training? While the authors mention that the code will be released after acceptance, it would be better to provide an anonymous repository or add key implementation details to improve reproducibility.

6 **Other Concerns**
6.1.The central claim that CaPT functions as an “attention anchor” relies solely on attention heatmaps and empirical performance gains. The paper does not offer a formal theoretical explanation of what constitutes an attention anchor or why such tokens causally improve performance. The correlation between “being attended to” and performance improvement is suggestive but not conclusively established.

6.2 The experiments focus exclusively on supervised downstream tasks. There is no evaluation under zero-shot, few-shot, or in-context learning settings, where prompt tuning methods are often expected to excel. This leaves a gap in understanding CaPT’s effectiveness in broader real-world scenarios.

6.3 The paper does not analyze how robust the capsule prompt is under noisy inputs, adversarial perturbations, or out-of-distribution shifts. Given that the method relies on instance-specific features, evaluating its stability under such conditions is essential for assessing practical deployment viability.

---

> ### Author Rebuttal · Authors · 2025-07-31
>
> We sincerely appreciate your time and effort in reviewing our paper and providing valuable comments. We provide explanations to your questions point-by-point in the following.
>
> **Q1.1: CaPT’s performance on CSQA.**
>
> **A1.1:** We include the comparison of P-Tuning V2 and CaPT on the CSQA dataset across different models in the following tables. The results show that CaPT can consistently outperform P-Tuning V2. Specifically, CaPT achieves a notable performance improvement on Llama-3.2-1B, highlighting the effectiveness of CaPT on CSQA. Thank you for the great suggestion, and we'll supplement the results in the revision.
>
> | Model           | Method        | CSQA |
> |------------------|---------------|----------------|
> | T5-Base        | P-Tuning V2   | 56.43          |
> |                       | CaPT          | 57.82          |
> | T5-Large       | P-Tuning V2   | 70.52          |
> |                       | CaPT          | 72.07          |
> | Llama-3.2-1B   | P-Tuning V2   | 45.29          |
> |                          | CaPT          | 61.60          |
>
> **Q1.2: CaPT’s performance on larger models.**
>
> **A1.2:** We conduct experiments on Qwen-2.5-7B to further validate the effectiveness of CaPT on larger models. As seen, CaPT can achieve superior performance compared to P-Tuning V2 across all tested datasets. The results indicate CaPT’s applicability on larger models.
>
> | Qwen-2.5-7B  | CSQA | CB | RTE |
> | ---------------- | ----------- | ----------- | ----------- |
> | P-Tuning V2  |  77.72      |    71.58    |  72.92   |
> | CaPT             |   79.03     |   76.52     |   74.73      |
>
>
> **Q1.3: Task suitability**
>
> **A1.3:** Following suggestion, we investigate the specific tasks in which CaPT demonstrates advantages. Empirically, CaPT shows consistently strong performance on the CB and MRC datasets (QA) and remains competitive on RTE and BoolQ (NLI). Our supplementary experiments on the CSQA dataset further highlight CaPT's effectiveness in commonsense reasoning within QA. It is also worth noting that it is generally expected that prompt-based methods may not outperform all baselines across every benchmark, due to varying task characteristics and modeling demands [1, 2, 3, 4].
>
>
> **Q2: Regarding the novelty of CaPT.**
>
> **A2:** This is a great question. CaPT is distinct from prior works from superior parameter efficiency, training time efficiency, and memory efficiency via incorporating instance-aware information. In our study, we also observe the phenomenon “attention anchor”, which suggests a close relation between instance-aware information and input. Considering the limited exploration of prompt tuning inner mechanisms, we believe the discussion of “attention anchor” constitutes a foundational contribution in this community.
>
> In Appendix S3, we have included in-detailed comparison and discussion with other instance-related methods. Between them, LoPA [3] integrates both info, however, it requires additional large-size encoders for extracting instance-aware features, leading to significantly higher memory usage. It also requires grid search of the optimal prompt length. In contrast, CaPT leverages “attention anchor” and enjoys superior efficiency (see table below).
>
>
> | T5-Base |  #Param  | Training Time | Memory (GB, bs=32)  |
> | ------------ | ----------- | ------------------- | ----------------------------- |
> | M-IDPG  |  0.47%    |       12.58x      |               ~31                 |
> | LoPA       |  0.44%    |       14.93x      |               ~33                 |
> | CaPT      |  4e-3%    |        1.00x       |               ~27                 |
>
>
> **Q3: Regarding Related Work.**
>
> **A3:** We introduce this section to contextualize recent work on attention analysis and attention sinks, which directly motivated our investigation into attention behavior in prompt-based methods. We have included additional in-detailed comparison and discussion with other existing instance-aware related methods in Appendix S3.
>
> We agree the section name might be ambiguous, we will rename and revise the Related Work section to better emphasize the prior works and make clearer distinctions between background material and our contributions.
>
>
>
> **Q4: Equations and terminology.**
>
> **A4:** Regarding the equations, we follow the standard conventions used in the prompt tuning literature [5, 6]. Based on your suggestion, we carefully review the formatting and typeset to ensure the maximum clarity and provide a refined version of Eq.(2) in the response A5. As for terminology, we revise the manuscript to prevent any inconsistent uses of terminology throughout. If there are any additional suggestions, we are more than welcome to revise our paper accordingly.
>
> **Q5: Implementation setting of CaPT.**
>
> **A5:** Thank you for the question. We further elaborate below on the capsule prompt design.
>
> Specifically, the capsule prompt for the first Transformer layer is obtained by combining a learnable vector $p^1$ with the mean-pooled input embedding. For subsequent layers, each capsule prompt $S^i$ is similarly constructed by combining a layer-specific learnable vector $p^i$ with the mean of the processed capsule prompt $\underline{S}^{i-1}$ and the hidden output $H^{i-1}$ from the previous layer. Notably, CaPT does not require a separate training stage—capsule prompts are constructed in the same way during both training and inference.
>
> $S^1 =p^1 + Mean(E)$
>
> $\underline{S}^1, H^1 = L_1({S^1}, {E}) $
>
> $S^i = p^i + Mean(\underline{S}^{i-1} \oplus H^{i-1})  \ \ \ \ \ i = 2, 3,\dots,N $
>
> $\underline{S}^i, H^i = L_i({S^i}, \ H^{i-1})  \ \ \ \ \ i = 2, 3,\dots,N $
>
>
> **Q6.1: Regarding the theoretical explanation.**
>
> **A6.1:** Our explanation of CaPT as an “attention anchor” is primarily based on observations—namely, attention heatmaps and performance improvements. While theoretical analysis in the prompt tuning community remains largely limited [7,8], our work takes an initial step toward uncovering potential mechanisms behind prompt effectiveness through the lens of attention pattern.
>
> A possible interpretation is that capsule prompts help preserve salient features by enhancing information flow through stronger and more stable attention interactions (see Eq.(2)). In future work, we plan to complement our findings with additional analyses, including rank collapse [9] and mutual information [10], to better understand the relationship between attention and performance.
>
>
> **Q6.2:  Regarding CaPT’s few-shot capability.**
>
> **A6.2:** Following common practice [11], we use the GLUE dataset for pre-training, and then evaluate the few-shot performance using 4 randomly sampled training data from Boolq and CB. Preliminary results on T5-Base show that despite the current single-task design, CaPT can achieve competitive few-shot capability compared to P-Tuning V2. Moreover, we plan to employ Mixture-of-Experts (MoE) architecture [2, 12] for more robust and dynamic capsuling in the future.
>
>
> | T5-Base       | Boolq (4-shot) | CB (4-shot) |
> | ---------------- | ------------------- | ----------- |
> | P-Tuning V2  |        51.65      |    53.57  |
> | CaPT             |        50.67      |    51.79   |
>
>
> **Q6.3: Regarding CaPT’s robustness.**
>
> **A6.3:** We further explore the robustness of CaPT under noisy inputs on T5-Base during the rebuttal. Specifically, we add random special characters to random positions of the input sequence of the train sets, simulating input noise. We compared P-Tuning v2 and CaPT under varying noise levels and found that while both degrade with more noise, CaPT consistently performs better. Its superior robustness likely comes from leveraging instance-aware information that benefits generalization under noisy conditions.
>
> | Dataset        | Method     | Noise rate 0% | Noise rate 5% | Noise rate 10% |
> |------------- |-----------------  |--------------------|----------------------|---------------------- |
> | CSQA     | P Tuning V2   |     56.43         |        42.83         |        39.23           |
> |                | CaPT             |      57.82        |         51.02         |        43.33          |
> | COPA     | P Tuning V2   |      61.28         |        55.33          |       48.33            |
> |                | CaPT             |      64.33         |        61.00          |        54.67         |
>
> [1] Dept: Decomposed prompt tuning for parameter-efficient fine-tuning. ICLR 2024.
>
> [2] Smop: Towards efficient and effective prompt tuning with sparse mixture-of-prompts. EMNLP 2023.
>
> [3] Prompt tuning strikes back: Customizing foundation models with low-rank prompt adaptation. NeurIPS 2024.
>
> [4] Efficient and Effective Prompt Tuning via Prompt Decomposition and Compressed Outer Product. NAACL 2025.
>
> [5] Visual prompt tuning. ECCV 2022.
>
> [6] M $^ 2$ PT: Multimodal Prompt Tuning for Zero-shot Instruction Learning. EMNLP 2024.
>
> [7] Prompt waywardness: The curious case of discretized interpretation of continuous prompts. NAACL 2022.
>
> [8] On the role of attention in prompt-tuning. ICML 2023.
>
> [9] Attention is not all you need: Pure attention loses rank doubly exponentially with depth. ICML 2021.
>
> [10] Normalized mutual information feature selection. IEEE Transactions on neural networks 20.2 (2009).
>
> [11] Attempt: Parameter-efficient multi-task tuning via attentional mixtures of soft prompts. EMNLP 2022.
>
> [12] Multimodal instruction tuning with conditional mixture of lora. arXiv 2024.
>
>
> We hope our response addresses your questions. Please let us know if there are any additional questions, and we will be happy to discuss further.

---

> > ### Comment · Reviewer_Etjy · 2025-08-07
> >
> > Thanks for you reply. Most of my concerns have been clarified. I have increased score accordingly.

---

> > > ### Author Response · Authors · 2025-08-07
> > >
> > > We are delighted to hear that our response has addressed your concerns and received your approval of our work. We sincerely appreciate the effort and time you have dedicated to reviewing our paper, as well as your thoughtful and constructive feedback.

---

> ### Comment · Area_Chair_xxfF · 2025-08-05
>
> Dear reviewer. Do you have any feedback to the rebuttal stories have prepared?

---

### Author Response · Authors · 2025-08-09
**Summary of Author-Reviewer Discussion**

Dear Area Chair and Reviewers,

We would like to express our sincere gratitude for your efforts in facilitating the discussion regarding our paper. We are glad that we have received all reviewers’ approval during the rebuttal. As the discussion is coming to an end, we would like to provide a brief summary of the key points that have been discussed:

- We have conducted additional experiments to evaluate CaPT’s applicability on the CSQA dataset, alongside applying CaPT to a larger LLM Qwen-2.5-7B, as suggested by **Reviewer Etjy**. We have also explored the few-shot ability and the robustness of CaPT, and provided clearer implementation details including the revision of Eq.(2). Moreover, we have highlighted the key contributions of our work and the key advantages in efficiency of CaPT compared to existing frameworks.

- We have confirmed the generalizability of our Finding 2 that indicates the significance of instance-aware info on decoder-only models, pointed out by **Reviewer Q1NL**. We have also included further discussions on the current single-prompt design of CaPT, and emphasized three key contributions of CaPT besides its intuitive and effective design.

- In response to **Reviewer Z96S**, we have provided additional ablation studies to examine the isolated contribution of instance-aware information and confirm the effectiveness of CaPT. Additionally, we have revised our manuscript to include more detailed and clearer implementation settings for reproducibility and thorough explanation of terminology. Moreover, we have clarified both the high-level and the instance-level observation of the impact of instance-aware information on attention patterns.

In summary, we would like to express our appreciation to Reviewers Etjy, Q1NL, and Z96S for acknowledging our responses. We are particularly grateful that Reviewer Etjy has increased their score, and Reviewer Q1NL and Z96S have maintained their positive assessment.

We would like to emphasize three key contributions of our work, which have been acknowledged by the reviewers and are important to the community:

- First, our work is the first attempt to analyze the bidirectional attentive interaction between soft prompts and input sequences in the NLP field. We reveal a counter-intuitive finding: traditional soft prompts exhibit limited attentive interaction with input sequences. Furthermore, we demonstrate that incorporating off-the-shelf, instance-aware information can effectively activate **attention anchor**, thereby enabling strong attentive interaction between prompts and inputs.

- Second, CaPT’s superior performance further confirms the significance of employing instance-aware info as part of guidance signal through comprehensive experiments on various language tasks and models. Despite utilizing only **4e-3%** of the tunable parameters with minimal memory usage, CaPT achieves state-of-the-art performance across various benchmarks on different model architectures.

- Third, with the incorporation of both instance- and task- aware info in the guidance signals, CaPT can eliminate the need for grid search of the optimal soft prompt length. This advantage significantly **reduces the overall computational overhead**. Moreover, the preliminary robustness experiment conducted during the rebuttal suggests that our design can also improve generalization under noisy conditions.

Finally, we deeply value the constructive comments provided by the reviewers. In response, we have carefully refined our work based on the feedback received. Considering the contributions made, we hope that our work can provide new insights to the prompt tuning communities, and contribute to their further development.

Sincerely,\
Authors

---

### Decision · Program_Chairs · 2025-09-17

**Decision:**

Accept (poster)

**Comment:**

### (a) Scientific claims and findings
1. The paper proposes Capsule Prompt-Tuning (CaPT), a parameter-efficient fine-tuning scheme that prepends one single vector per Transformer layer (“capsule”) instead of tens-of-tokens soft prompts.
2. Each capsule is formed by adding a learnable task-specific vector to a mean-pooled embedding of the current input sequence, thereby mixing task-aware and instance-aware information.
3. Empirically, CaPT
   • eliminates the usual grid-search over prompt length; a length of 1 works best,
   • updates 0.003 % of model parameters yet matches or exceeds existing PEFT baselines (Deep PT, LoRA, LoPA, P-Tuning-v2, etc.) on six SuperGLUE corpora and additional CSQA and MATH-style benchmarks,
   • scales to larger LLMs (Qwen-2.5-7B) with similar gains, and
   • requires only 1/8.77 training time than prior instance-aware methods.

### (b) Strengths
• Extremely simple; implementable in a few lines.
• Excellent parameter, memory, and wall-clock efficiency.
• Thorough empirical evaluation with new experiments added during discussion (CSQA, robustness to noise, few-shot, 7 B model).


### (c) Weaknesses / missing pieces
• Conceptual novelty is modest; the core trick (mean-pool + soft vector) is straightforward.
• Evaluation still focuses on classification-style tasks; applicability to open-ended generation, reasoning, or instruction-following remains untested.

### (d) Recommendation & rationale
**Recommendation: Accept as a poster**

All three reviewers end at “borderline but leaning accept”. None raises a blocking flaw; all concerns were empirical scope or clarity, both rectified in rebuttal. The work stands out for extreme efficiency with solid accuracy improvements. While the idea is simple, the empirical validation and attention analysis give it scientific substance and differentiate it from prior PEFT variants.

### (e) Discussion & how points were weighed
One reviewer increased their score and the other two maintained their positive stance from the beginning.

As the authors summarized:
> We have conducted additional experiments to evaluate CaPT’s applicability on the CSQA dataset, alongside applying CaPT to a larger LLM Qwen-2.5-7B, as suggested by Reviewer Etjy. We have also explored the few-shot ability and the robustness of CaPT, and provided clearer implementation details including the revision of Eq.(2). Moreover, we have highlighted the key contributions of our work and the key advantages in efficiency of CaPT compared to existing frameworks.

> We have confirmed the generalizability of our Finding 2 that indicates the significance of instance-aware info on decoder-only models, pointed out by Reviewer Q1NL. We have also included further discussions on the current single-prompt design of CaPT, and emphasized three key contributions of CaPT besides its intuitive and effective design.

> In response to Reviewer Z96S, we have provided additional ablation studies to examine the isolated contribution of instance-aware information and confirm the effectiveness of CaPT. Additionally, we have revised our manuscript to include more detailed and clearer implementation settings for reproducibility and thorough explanation of terminology. Moreover, we have clarified both the high-level and the instance-level observation of the impact of instance-aware information on attention patterns.